# Genomic Characterization of the Titan-like Cell Producing *Naganishia tulchinskyi*, the First Novel Eukaryote Isolated from the International Space Station

**DOI:** 10.3390/jof8020165

**Published:** 2022-02-08

**Authors:** Swati Bijlani, Ceth Parker, Nitin K. Singh, Maria A. Sierra, Jonathan Foox, Clay C. C. Wang, Christopher E. Mason, Kasthuri Venkateswaran

**Affiliations:** 1Department of Pharmacology and Pharmaceutical Sciences, School of Pharmacy, University of Southern California, Los Angeles, CA 90089, USA; bijlani.swati@gmail.com (S.B.); clayw@usc.edu (C.C.C.W.); 2Biotechnology and Planetary Protection Group, Jet Propulsion Laboratory, California Institute of Technology, Pasadena, CA 91109, USA; Ceth.w.parker@jpl.nasa.gov (C.P.); Nitin.K.Singh@jpl.nasa.gov (N.K.S.); 3Institute for Computational Biomedicine, Weill Cornell Medicine, New York, NY 10021, USA; mas4037@med.cornell.edu; 4Tri-Institutional Computational Biology & Medicine Program, Weill Cornell Medicine, New York, NY 10021, USA; jof3004@med.cornell.edu; 5The WorldQuant Initiative for Quantitative Prediction, Weill Cornell Medicine, New York, NY 10021, USA

**Keywords:** *Naganisha tulchinskyi*, yeast, International Space Station, phylogenetic analyses, Titan-like cells, simulated microgravity

## Abstract

Multiple strains of a novel yeast belonging to genus *Naganishia* were isolated from environmental surfaces aboard the International Space Station (ISS). These strains exhibited a phenotype similar to Titan cell (~10 µm diameter) morphology when grown under a combination of simulated microgravity and 5% CO_2_ conditions. Confocal, scanning, and transmission electron microscopy revealed distinct morphological differences between the microgravity-grown cells and the standard Earth gravity-grown cells, including larger cells and thicker cell walls, altered intracellular morphology, modifications to extracellular fimbriae, budding, and the shedding of bud scars. Phylogenetic analyses via multi-locus sequence typing indicated that these ISS strains represented a single species in the genus *Naganishia* and were clustered with *Naganishia diffluens*. The name *Naganishia tulchinskyi* is proposed to accommodate these strains, with IF6SW-B1^T^ as the holotype. The gene ontologies were assigned to the cell morphogenesis, microtubule-based response, and response to UV light, suggesting a variety of phenotypes that are well suited to respond to microgravity and radiation. Genomic analyses also indicated that the extracellular region, outer membrane, and cell wall were among the highest cellular component results, thus implying a set of genes associated with Titan-like cell plasticity. Finally, the highest molecular function matches included cytoskeletal motor activity, microtubule motor activity, and nuclear export signal receptor activity.

## 1. Introduction

The genus *Naganishia* [1] was resurrected from *Cryptococcus* and emended to accommodate the *albidus* clade of *Filobasidiales*, Tremellomycetes, and Basidiomycota [2,3,4]. The genus *Naganishia* contains the following species as legitimate in MycoBank: *N. adeliensis, N. albida, N. albidosimilis, N. antarctica, N. bhutanensis, N. brisbanensis, N. cerealis, N. diffluens, N. friedmannii, N. globosa, N. indica, N. liquefaciens, N. nivalis, N. onofrii, N. qatarensis, N. randhawae, N. uzbekistanensis, N. vaughanmartiniae*, and *N. vishniacii* [2,3,4]. However, the addition of *N. floricola* made the list a total of 20 *Naganishia* species, but several of these members were yet to be validly described. In an ongoing Microbial Tracking experiment on the International Space Station (ISS), six yeast strains belonging to the family *Tremellomycetes* were isolated [5], and whole-genome sequences (WGS) were generated [6]. Some of the *Naganishia* species that were phylogenetically related to these ISS strains, which were reported to be psychrophilic or psychrotolerant species, isolated from Antarctica [4,7,8,9,10,11], also reported that the ISS *Naganishia* species might also display properties related to cold adaptation and an increase in radiation resistance. In addition, *Naganishia* species were also isolated from a variety of ecosystems, such as water [12], soil [13], tree trunk [14], air [12], Alpine glaciers [15], cheese, cereals [16], and human specimens [12].

Members of the *Cryptococcus* species, from which *Naganishia* species have recently been separated, were reported to possess unique virulence factors associated with the pathogenesis [17,18]. The morphological transformation of normal *Cryptococcus* cells into giant (Titan) cells was one of the factors contributing to cryptococcosis (an inhalation acquired infection) [19,20,21]. It was also reported that titanization was unique to the *Cryptococcus neoformans/C. gattii* species complex, and not found in other budding basidiomycetous yeasts or in the ascomycetous yeast *Saccharomyces cerevisiae*, when grown under similar growth conditions [21]. However, occasional enlarged cells have been reported in *Cryptococcus terreus*. The ISS *Naganishia* strains exhibited Titan-like cells, a phenotype similar to the *C. neoformans/C. gattii* species complex, when the ISS isolates were grown under simulated microgravity conditions.

One of the objectives of this study was to characterize the Titan-like cells of the novel *Naganishia* species isolated from the ISS environment. The microscopy analyses revealed the presence of Titan-like giant cells with abnormal morphology when *Naganishia* species were subjected to several stressors, including a combination of hypoxic 5% CO_2_ and simulated ISS microgravity conditions. The second objective of this study was to utilize WGS for annotation in order to aid in the identification of genetic determinants related to cold adaptation and radiation resistance in *N. tulchinskyi*. The third objective was to determine the phylogenetic novelty of the ISS *Naganishia* strains (*n* = 4) using microscopy and taxonomic affiliation based on multi-locus sequence typing (MLST) analyses, which included a seven gene concatenated (ITS, LSU, SSU, *CYTB*, *RPB1*, *RPB2*, *TEF1*) phylogeny [2].

## 2. Materials and Methods

### 2.1. Sample Collection and Isolation of Microorganisms

During the flight experiments of Microbial Tracking-1, several surface samples (1 m^2^) were collected from various ISS environmental locations. Previous publications had detailed sample collection, processing, and isolation of cultivable microorganisms from the ISS [5]. Briefly, internal ISS surface samples were collected using polyester wipes and then transported to Earth for processing. Samples were collected just prior to departure (within 5 days) and stored at 4 °C in the ISS, transported in cold packs (~4 °C) from ISS to the Earth, hand delivered at Long Beach, California after splash down, which were then carried in ice packs (~4 °C) during ground transportation, before storing at JPL at 4 °C for further processing. Total number of days for sample storage time from sample collection to processing was ~7 days. Unless otherwise stated, all reagents were purchased from Sigma Aldrich (St. Louis, MO, USA). The materials associated with the sampling wipes were placed in 200 mL of sterile phosphate-buffered saline (PBS; pH 7.4) solution and plated onto potato dextrose agar (PDA), supplemented with chloramphenicol (100 µg/mL; *v*/*v*), to isolate fungi [5,22]. Microbial colonies grown at 25 °C for 7 days were picked from the media plates, purified, and stored for further analyses. Based on internal transcribed spacer (ITS) sequencing, six yeast strains were tentatively assigned to the genus *Cryptococcus*, and subsequent WGS characterization revealed that two strains were identified as *Papiliotrema laurentii*, while four strains belonged to a yet to be described *Naganishia* species [6], making this species the first known novel eukaryote to be isolated from the ISS [6].

### 2.2. Whole Genome Sequencing Analyses

The WGS used for the analyses were previously generated and published [6]. The WGS raw data was filtered with NGS QC Toolkit v2.3 [23] for high-quality (HQ) vector and adaptor-free reads for genome assembly (cutoff read length for HQ, 80%; cutoff quality score, 20). The number of filtered reads obtained were used for assembly with SPAdes 3.14.0 [24] genome assembler (k-mer size—32 to 72 bases) using default parameters. Genomic DNA sequences and predicted proteomes (where available) of all other strains used in this study were downloaded from NCBI, followed by comparative analyses.

### 2.3. Phylogenetic Analyses

Among the 20 known *Naganishia* species, NCBI had ITS sequences available for all the species except *N. indica*. The gene sequences of all available strains were retrieved from NCBI, except for the four ISS strains, which were derived from their WGS during this study. Phylogenetic analyses based on individual genes (ITS, LSU, SSU, *CYTB*, *RPB1*, *RPB2*, *TEF1*) and MLST analyses (with the seven genes) were carried out with all available *Naganishia* species (*n* = 10). The individual gene sequences for all strains were first aligned separately using ClustalW, and for MLST, seven gene sequences for each strain were concatenated manually, aligned using ClusatalW, and used to generate a Maximum Likelihood Tree using MEGA 7.0.26 [25].

### 2.4. Morphological Characterization under Simulated Microgravity Conditions

#### 2.4.1. Growth Conditions for Phenotypic Analyses under Different Conditions

*Naganishia* sp. IF6SW-B1^T^ was streaked on a Yeast extract- Peptone- Dextrose agar (1% yeast extract, 2% peptone, 2% glucose, and 1.5% agar) plate and incubated at 30 °C for 48 h. After 48 h of growth, colonies were resuspended in PBS, and the cells were enumerated using a hemocytometer. The cells were then inoculated at an initial density of 2 × 10^5^ cells/mL in synthetic dextrose (SD) broth. Replicates (*n* = 3) of each yeast strain were grown under each condition tested. The cells were inoculated in three conditions; (a) the first set was inoculated in 5 mL of liquid media in a tube, incubated at 30 °C, and shaken at 150 rpm; (b) the second set was inoculated in 5 mL liquid media and incubated on a rotating platform in a CO_2_ incubator (5% CO_2_) at 30 °C; (c) the final set was grown in 10 mL of liquid media in a high aspect ratio vessels (HARV, Synthecon Inc., Houston, TX, USA) and mounted onto a vertical rotating device (at 30 rpm) to simulate microgravity conditions, these were placed in a CO_2_ incubator (5% CO_2_) at 30 °C. The CO_2_ level of ISS habitat is ~10× more (~4000 ppm) than at Earth conditions [5]. Additionally, Titan cells were observed under 5% CO_2_ level in *C. neoformans* [26]. Hence, morphological changes were characterized at this 5% CO_2_ level. The autoclavable HARVs provided oxygenation via a flat, silicone rubber gas transfer membrane (Synthecon Inc.). The cultures were all incubated as described for 4 days, followed by cell pelleting for further analyses.

The biochemical and physiological characterizations were characterized by employing BioLog Yeast plates, as described elsewhere [27]. Briefly, 4-days grown colonies in PDA plates were collected in sterile distilled water aseptically, and ~10^6^ cells were transferred into the BioLog Yeast plates (BioLog Inc., Hayward, CA, USA) and incubated at 25 °C for 7 days. The change in coloration and oxidation/assimilation profiles of 94 different carbon sources were documented by visual examination. The clear positive and negatives were scored in 7 days incubation, and delayed reaction (weak) was documented by examining the tetrazolium violet color change after 14 days incubation. Further incubation and observation after 30 days period of time did not yield any difference in results compared to those that were observed at day 7.

#### 2.4.2. Confocal Microscopy

Yeast cell samples grown under the three conditions described above were fixed in 4% paraformaldehyde (PFA) in PBS for 15 min at 4 °C. These cells were gently pelleted by centrifugation for 5 min at 3500 rpm, and cells were rinsed 3× with sterile PBS. Subsequent processing steps were carried out in the dark, where samples were covered with aluminum foil and protected from the light. Cell pellets were resuspended in 15 µM Calcofluor White and incubated for 1 h at 37 °C. Samples were then rinsed and pelleted as previously mentioned but with distilled H_2_O. After the third rinse, samples were pelleted again, and the pellet was resuspended with 1 µM TO-PRO^®^-3 (Fisher Scientific, Los Angeles, CA, USA) and incubated for 30 min at room temperature. Samples were further rinsed and pelleted, as previously mentioned, using distilled H_2_O. The resuspended samples were filtered through a 0.2 µM 25 mm Isopore Membrane Filter (GTBP) using a filter manifold. This membrane was removed, placed on a microscope slide, and a drop of antifade was added to the top. A coverslip was placed, and then, the edges were sealed. The samples were viewed using a Zeiss LSM 880 upright laser scanning confocal microscope and an Alpha Plan-APOCHROMAT 63× lens (Zeiss, Oberkochen, Germany). Yeast cell size and cell wall thickness were analyzed using ImageJ [28].

#### 2.4.3. Scanning Electron Microscopy

Yeast cells grown under various conditions were preserved using 4% paraformaldehyde (PFA) in PBS for scanning electron microscopy (SEM) and transmission electron microscopy (TEM). Following the concentration, cells were immersed in a chilled 4% PFA and incubated at 4 °C for 10 min before the PFA was removed and washed in PBS buffer. Fixed cells were transferred into a 24-well plate, dehydrated in an isopropyl alcohol (IPA) series (50% to 100%), and stored at 4 °C. Samples were critically point dried in an Automegasamdri 915B critical point dryer (Tousimis, Rockville, MD, USA). Samples were attached to SEM stubs with carbon tape (Ted Pella Inc., Redding, CA, USA), followed by carbon coating with a Leica EM ACE600 Carbon Evaporator (Leica, Wetzlar, Germany) to a thickness of ~12 nm. SEM analysis was performed with an FEI Quanta 200F (Thermo Fisher, Waltham, MA, USA).

#### 2.4.4. Transmission Electron Microscopy

TEM was performed to examine cell wall thickness variations and analyze organelles’ differences between the different environmental stressors (CO_2_ and simulated microgravity) induced Titan-like cells and standard cells, grown under Earth gravity, with 5% CO_2_ and without CO_2_. For TEM, yeast cells were fixed in 2% glutaraldehyde for at least one hour at 4 °C. Samples were rinsed in 0.1 M sodium cacodylate buffer, pH 7.2, and immersed in 1% osmium tetroxide with 1.6% potassium ferricyanide in 0.1 M sodium cacodylate buffer for 1 h. After rinsing, samples were polymerized at 60 °C for 24–48 h in silicon molds. Thick sections (200 nm) were cut, stained with toluidine blue, and used. Thin sections (70 nm) were collected onto formvar-coated 50 mesh copper grids. The grids were post-stained with 2% uranyl acetate, followed by Reynold’s lead citrate. The sections were imaged using a Tecnai 12 120 kV TEM (FEI, Hillsboro, OR, USA) and data were recorded using an UltraScan 1000 with Digital Micrograph 3 software (Gatan Inc., Pleasanton, CA, USA).

#### 2.4.5. Statistical Analyses

All statistical analyses were performed using GraphPad Prism version 9.1.0 (GraphPad Software, San Diego, CA, USA). Specifically, data was tested for normal distribution with the D’Agostino and Pearson normality test, followed by non-parametric ANOVA using a post-hoc Kruskal–Wallis analysis.

### 2.5. Survey of Genes and Proteins Involved in Resistance

Genes and proteins involved in cold resistance from the closely related species *N. vishniacii* [29] were screened using tBLASTn and BLASTp (Table 1). Cold tolerance included genes involved in carotenoid synthesis: isopentenyl diphosphate isomerase, farnesyltransferase, farnesyl pyrophosphatase synthetase, phytoene-beta carotene synthase, NADP-cytochrome P450 reductase; genes involved in mycosporine synthesis: 2-epi-5-epi-valiolone synthase, catechol o-methyltransferase, and carbamoylphosphate synthase reported in *Phaffia rhodozyma* [30]; enzymes involved in the formation and degradation of trehalose: trehalose-6-phosphate synthase component TPS1 and TPS2, alpha, alpha-trehalase NTH2, NTH1, and ATH1. Additionally, resistance related proteins for fungi were screened using sequences obtained from NCBI by using ‘microgravity’ and ‘radioresistance’ keywords. Microgravity-related proteins included: Daughter-specific expression protein 1 and 2, GPI-anchored protein 24, Chain A Flocculation protein FLO1, Chain A and B Flocculin, and Chain A proteinase K. Radioresistance proteins including sequences from the DNA repair protein rad9.

### 2.6. De Novo Genome Annotation

Gene annotation on each of the six assemblies (four *Naganishia tuchinskyi* strains and two closely related species) was performed using MAKER [31]. MAKER identified repeats, aligned expressed sequence tags (ESTs) and proteins to a genome, and produced ab initio gene predictions. A total of 74,830 available ESTs from closely related *Naganishia* species were downloaded from NCBI (Table 2) and protein sequences from six strains were downloaded from UniProt [32].

Firstly, de novo repeat identification and masking were performed using RepeatModeler [33]. Then, MAKER control files were generated (maker-CTL), and a first-round was run using the maker_opts.ctl control file. Data for the transcriptome assembly (EST), protein sequences (protein), and repeat annotations (rm_gff) was provided. Additionally, the model_org was set to ‘simple’ to annotate simple repeats, and inference predictions of genes directly from ESTs and protein homology were set (est2genome = 1, protein2genome = 1). Lastly, GFF and FASTA outputs were assembled together with GFF3toolkit [34].

Next, we predicted genes using the gene models generated with MAKER by training SNAP [35] and Augustus [36] using BUSCO [37]. For SNAP, we used models with an AED of 0.25 or better and a length of 50 or more amino acids. To train Augustus, we excised regions that contain mRNA annotations based on our initial MAKER run, with 1000 bp on each side, as described [38].

To run BUSCO, the Eukaryota set of conserved genes was used. The -m genome option was specified to provide regions that included more than just transcripts. The initial HMM model selected was *Cryptococcus*, which corresponded to the closest genus of *Naganishia* (augustus_species = cryptococcus). The long option was set to optimize self-training Augustus.

Following the BUSCO run, the second round of MAKER was performed using the SNAP and Augustus models. For this purpose, FASTA sequence files were replaced with GFF files (est_gff, protein_gff, and rm_gff), and est2genome and protein2genome were switched to 0. The third run of MAKER was performed, and the gene models were evaluated by counting the number of gene models and the gene lengths after each round and running BUSCO using the transcript sequences (-m transcriptome).

**Table 1 jof-08-00165-t001:** Presence of various biomolecules responsible for pigment biosynthesis, radio-resistance, and microgravity tolerance in *Naganishia* species and *S. cerevisiae*.

Functions	GenBank#	Enzyme, Protein, Other Biomolecules	*Naganishia tulchinskyi*				*Naganishia vishniacii*	*Papiliotrema laurentii*		*S. cerevisiae*	Reference
			IF6SW-B1^T^	1F7SW-B1	1F1SW-F1	IIF5SW-F1	CBS 10616	IF7SW-B5	IF7SW-F4	S288C	
Pigment Biosynthesis	AAY33922	Geranylgeranyl diphosphate synthase	*	**	**	**	**	*	**	**	[39]
	ACI43098	Cytochrome P450 reductase	**	**	**	**	**	**	**	**	[40]
	AHW57996	Farnesyl pyrophosphate synthase	**	**	**	**	**	**	**	**	[41]
	BAA33979	Isopentenyl-diphosphate delta isomerase	**	**	**	**	**	**	**	**	[42]
	NP_009555	Alpha, alpha-trehalase NTH2	**	**	**	**	**	**	**	**	[43]
	NP_010284	Alpha, alpha-trehalase NTH1	**	**	**	**	**	**	**	**	[44]
	NP_015351	Alpha, alpha-trehalase ATH1								**	[43]
	NP_594975	Putative trehalose-phosphate synthase Tps2	**	**	*	**	**	**	**	**	[45]
	AAO47570	Phytoene-beta carotene synthase	**	**	**	**	*	*	*		[46]
Radio- resistance	RAD51	Recombinase RAD51	**	**	**	**	**	**	**	**	[47]
	RAD54	DNA repair and recombination protein RAD54-like protein	**	**	**	**	**	**	**	**	[47]
	RDH54	DNA-dependent ATPase RDH54	**	**	**	**	**	**	**	**	[47]
	CAA54491	RAD9						*	*		[48]
	CAA54492	RAD9							*		[48]
Microgravity tolerance	1HT3_A	Chain A, Proteinase K	**	**	**	**	**	**	**	**	[49]
	1IC6_A	Chain A, Proteinase K	**	**	**	**	**	**	**	**	[50]
	4LHK_A	Chain A, Flocculin								**	[51]
	4LHK_B	Chain B, Flocculin								**	[51]
	4LHN_A	Chain A, Flocculation protein FLO1								**	[51]
	P26306	DNA repair protein rad9						*	*		[52]
	P38844	Daughter-specific expression protein 2								**	[53]
	P40077	Daughter-specific expression protein 1								**	[54]
	P48013	DNA repair protein rad9							*		[48]
	Q59Y31	Flocculation protein 1								**	[55]

** Match > 75% of the protein sequence and e-value < 1 × 10^−25^; * Match < 75% of the protein sequence or e-value > 1 × 10^−25^.

**Table 2 jof-08-00165-t002:** Datasets used for De Novo gene annotation using MAKER.

Name	Origin *	# of Sequences	Date of Access	Data Set
*Naganishia tulchinskyi* (*n* = 4)	This study			Assembly
*Naganishia cerealis*	UniProt	3	4/5/21	Protein sequences
*Naganishia diffluens*	UniProt	7	4/5/21	Protein sequences
*Naganishia liquefaciens*	UniProt	4	4/5/21	Protein sequences
*Naganishia* sp.	UniProt	3	4/5/21	Protein sequences
*Cryptococcus neoformans var grubii*	UniProt	65	4/5/21	Protein sequences
*Cryptococcus neoformans var neoformans*	UniProt	370	4/5/21	Protein sequences
*Cryptococcus neoformans var grubii*	ESTdb	69	3/31/21	Transcriptome
*Cryptococcus neoformans var neoformans*	ESTdb	74,724	3/31/21	Transcriptome
*Cryptococcus vishniacci (Naganishia vishniacii)*	ESTdb	37	3/31/21	Transcriptome
*Papiliotrema laurentii* (*n* = 2)	This study			Assembly

* Expressed Sequence Tags database (ESTdb; Nature Genetics 4:332–3;1993) is a division of GenBank that contains sequence data and other information on “single-pass” cDNA sequences from a number of organisms. UniProt (2021) [32]. UniProt: the universal protein knowledgebase in 2021. Nucleic Acids Res 49, D480–d489.

### 2.7. Functional Annotation

Predicted protein sequences generated by MAKER were analyzed using InterProScan v5.51–85.0 [56] with default parameters. Functional annotations assigned by InterProScan per protein sequence were converted into collections of Gene Ontology (GO) terms using BLAST2GO [57] with default parameters. All assigned GO terms per protein were aggregated and visualized using the REVIGO web tool [58] with the “tiny” similarity cutoff (0.4) searched against the *Saccharomyces cerevisiae* S288C database and using the SimRel measurement of similarity. Visualizations were generated using the code provided by REVIGO and using R3.6.3.

### 2.8. Data Availability

The WGS and raw data have been deposited in GenBank under the BioProject accession number PRJNA623412 [6]. The genome sequences have also been deposited in the National Aeronautics and Space Administration (NASA) GeneLab system (GLDS-290; https://genelab-data.ndc.nasa.gov/genelab/accession/GLDS-290; accessed on 5 December 2021). In addition to the WGS, we performed amplicon sequencing the ITS region from each strain [5], and their accession numbers are given below. *Naganishia* IF6SW-B1 (KY218715.1); *Naganishia* IF1SW-F1 (KY218664.1); *Naganishia* IF7SW-B1 (KY218717.1); *Naganishia* IIF5SW-F1 (KY218695.1).

## 3. Results and Discussion

Several yeast strains belonging to the *Naganishia* (*Cryptococcus*) *diffluens* clade of *Filobasidiales* were isolated from two different flight collection experiments conducted aboard the ISS [5]. These strains, referred to as IF6SW-B1^T^, IF7SW-B1, IF1SW-F1, and IIF5SW-F1, were isolated from four different ISS locations and identified to be novel species belonging to the genus *Naganishia,* based on genomic characterization via polyphasic taxonomic approaches. The sampling of ISS surfaces, performed for this study, took place within the US on-orbit segments. Sampling campaign (denoted as I) was carried out on 4 March 2015, and the second one (denoted as II) was conducted on 15 May 2015. Samples collected during this study were: Node 3 (Locations #1, #2, and #3), Node 1 (Locations #4 and #5), Permanent Multipurpose Module (Location #6), U.S. Laboratory (Location #7), and Node 2 (Locations #8 and control). A detailed description of the various locations sampled was published elsewhere [59]. Strains IF6SW-B1, IF7SW-B1, and IF1SW-F1, were isolated during first sampling from Locations 6, 7, and 1, respectively. Strain IIF5SW-F1 was isolated from second sampling at Location 5.

### 3.1. Phylogenetic Analyses of Novel ISS Strains

To confirm that these four ISS strains form a novel species, their phylogenetic affiliations were analyzed with other species belonging to the genus *Naganishia*. The MycoBank database documented 19 *Naganishia* species, and the CBS database showed only 11 *Naganishia* species. The gene sequences available on NCBI for *Naganishia* species (*n* = 10) were used to generate phylogenetic maximum likelihood trees with *C. neoformans* CBS 132 strain as an outgroup.

Phylogenetic analyses were carried out using seven genes: internal transcribed spacer (ITS) region rRNA gene, D1/D2 domain of large subunit (LSU or 26S) rRNA gene, small subunit (SSU or 18S) rRNA gene, and housekeeping genes, including two subunits of RNA polymerase II (*RPB1* and *RPB2*), translation elongation factor 1-α (*TEF1*), and cytochrome b (*CYTB*), which were used for differentiating *Tremellomycetes* species [2]. The respective gene sequences that were available on NCBI for different *Naganishia* species were included in the phylogenetic analyses. In addition, genomes of closely related members from *Tremellomycetes* (*Goffeauzyma gastricus* CBS 2288 (AF145323.1), *Heterocephalacria arrabidensis* CBS 8678 (AF444362.2), *Piskurozyma cylindrica* CBS 8680 (AF444360.1), *Solicoccozyma aerius* CBS155 (AF145324.1), and *Tremella mesenterica* CBS 6973 (AF444433.1)) were included with *Cryptococcus neoformans* CBS 132 (AF444326.1) as the outgroup.

Initially, seven individual phylogenetic trees using ITS, LSU, SSU, *CTYB*, *RPB1*, *RPB2*, and *TEF1* genes were generated and are presented in Appendix A. When tested individually, the phylogenetic analyses generated based on individual gene, formed a single group for all four ISS strains. There were no significant variations in ITS, LSU, and SSU gene sequences (Appendix A), whereas the other four genes (*CYTB*, *RPB1*, *RPB2*, and *TEF1*) invariably exhibited higher resolution in differentiating all the *Naganishia* species. The ITS tree (Appendix A) showed that the ISS strains were placed at the same position of the three recognized species, *N. diffluens, N. liquefaciens* and *N. albidosimilis*. ITS sequences of the four ISS isolates showed 100% similarity to the sequence of *N. albidosimilis* CBS 7711^T^ (AF145325), *N. liquefaciens* CBS 968^T^ (AF444345), and *N. diffluens* CBS 160^T^ (AF145330). However, ITS sequences of *N. vishniacii* CBS 7110^T^ shared 97.84% similarity (13 bp difference to GenBank accession no. AF145320.), *N. uzbekistanensis* CBS 8683^T^ had 97.55% similarity (14 bp difference to GenBank accession no. AF444339), and *N. adeliensis* CBS 8351^T^ exhibited 97.34% similarity (14 bp difference to GenBank accession no. AF145328) with ISS isolates (Appendix A). In the LSU tree (Appendix A), the four ISS strains were placed at the same position that phylogenetically distinguished from *N. albida, N. adeliensis, N. vishniacii*, and *N. albidosimilis* with low bootstrap support. Comparing the LSU sequences of these ISS strains with closely related species showed a 97.71% similarity to *N. albida* (type strain, CBS 142; 14 different base pairs; GenBank accession no. AF075474), 96.41% similarity to *N. albidosimilis* (type strain, CBS 1926; 22 different base pairs; GenBank accession no. AF137601), 96.25% similarly to *N. liquefaciens* (type strain, CBS 968; 23 different base pairs; GenBank accession no. AF181515), and 95.92% similarity to *N. diffluens* (type strain, CBS 160; 25 different base pairs; GenBank accession no. AF075502) (Appendix A). The taxonomic threshold predicted to discriminate basidiomycetous yeast species is 99.51% in the LSU region and 99.21% in the ITS region [1]. Hence, the sequence data and phylogenetic analysis of the LSU regions of the ISS isolates confirm they belong to a new species in the genus *Naganishia* (order Filobasidiales). The SSU tree (Appendix A) demonstrated that these four strains were not separated from any of the recognized species. 

The other phylogenetic trees based on *CYTB*, *RPB1*, *RPB2*, and *TEF1* grouped the ISS strains together, and the closest neighbor was identified as *N. diffluens*. Subsequently, MLST was carried out by manually concatenating the seven genes (Figure 1). The MLST phylogenetic tree constructed showed the four ISS strains clustered together in the same clade with *N. diffluens* CBS 160^T^ (Figure 1). The individual gene-based and MLST tree-based analyses both further supported the finding that the four ISS strains belong to the same species which is closely related to, but separate from, *N. diffluens*. This suggested that the four ISS strains were novel species of the genus *Naganishia*. All four strains were isolated during different flights and sampling sessions at various locations, suggesting that these isolates are not clonal but persisting in the ISS environment. The ecological significance of this finding, in the closed system aboard the ISS warrants, further study. The genomes of four ISS strains were sequenced, draft genome was assembled, annotated, and the results were published elsewhere [6]. The genome size was ~19.4 Mbp with GC content between 53–56%, similar to other members of the genus *Naganishia*.

### 3.2. Titan-like Cell Characterization of Novel ISS Strains

IF6SW-B1^T^ strain was grown under culture conditions similar to those used for titanization of *C. neoformans*, and analyzed using a variety of microscopy techniques. Cell size and cell wall thickness were determined using Calcofluor White stain (Figure 2), Titan-like cell morphology using SEM (Figure 3), and cell wall structure using TEM (Figure 4). The Calcofluor White Stain (Figure 2A) binds with chitin contained within cell walls, and once triggered by its excitation wavelength (380 nm), the stain fluoresces and emits blue light at 475 nm. Bright-field images were captured alongside Calcofluor White images to show cell distribution and confirm successful Calcofluor White staining, while Calcofluor White micrographs were used to measure the size of the cells and their thickness (Figure 2A). Figure 2B shows the cell size in diameter (µm) under various growth conditions, whereas Figure 2C depicts the cell wall thickness. The size of the cells grown under standard Earth gravity conditions and in the presence of 5% CO_2_ with standard Earth gravity ranged from 2.5 µm to 6 µm; however, the cells grown under 5% CO_2_ were statistically significantly larger (*p* ≤ 0.05). When the strain was grown under 5% CO_2_ and simulated microgravity (µG) conditions, the cell size increased at least two to three times in size (4 µm to 13 µm; mean increased size of ~10 µm) and was statistically significantly larger than both the standard gravity control (*p* ≤ 0.0001) and the 5% CO_2_ treatment (*p* ≤ 0.0001) (Figure 2B). There was no statistical difference for cell wall thickness between the standard gravity control and the 5% CO_2_ grow cells (ns). However, the cell wall thickness was two to three-times larger when cells were grown under simulated microgravity conditions compared with the other two culture conditions (0.55 µm and 0.40–0.39 µm, respectively), these findings were statistically significant (*p* ≤ 0.0001) (Figure 2C).

The budding scars in IF6SW-B1^T^ strain seemed to exhibit normal bipolar budding pattern when grown under standard gravity (Figure 3A) and when grown and exposed to 5% CO_2_ with standard Earth gravity (Figure 3B). However, when the strain is grown under simulated microgravity and 5% CO_2_ conditions, multilateral budding is seen (Figure 3C,D). In addition, the budding cells were not as predominant under microgravity as those grown under normal gravity conditions (Figure 3A). Finally, when the IF6SW-B1^T^ strain was grown under 5% CO_2_, as well as simulated microgravity (Figure 3C,D) conditions, observations of budding cells were much less commonly observed, whereas the budding scars were evident and appeared to take up a larger proportional area on the yeast cell surface. The budding scars under simulated microgravity conditions range in morphology and size from 2–3 µm raised prominences extending 1–2 µm from the cell surface to large flat raised rings up to 5 µm in diameter (Figure 3C,D). Despite these morphological differences, microgravity grown yeast cells were viable when re-streaked onto a PDA plate. Comparatively, the number of viable cells was less when grown under 5% CO_2_ and simulated microgravity compared to the other standard atmosphere and gravity conditions (data not shown). These results suggested that the cells were defective in cytoskeleton formation, which might limit the formation of viable daughter cells. Cell wall structure analysis was then performed for the IF6SW-B1^T^ strain using TEM data. A simple comparison of the cell wall thickness of the cells grown under standard Earth gravity (Figure 4A,B) with simulated microgravity (Figure 4E, F) conditions exhibited enormous size (~5×) differences and substantiated the confocal microscopy-based measurements.

Comparison of confocal, SEM, and TEM (Figure 2, Figure 3 and Figure 4) between the three different conditions indicated that the µG grown Titan-like cells were much larger and exhibited irregular morphologies, including the novel discovery of peeling and shedding bud scars (Figure 3D, Figure 4E, and Appendix A). Indeed, a large majority of the disfigurement and abnormal cell morphology appeared to be associated with the surface location where daughter cell replication occurred (Figure 3C,D and Appendix A). Further analyses of TEM images emphasized the differences between the growth conditions, with µG grown cells showing large structural alterations within the intracellular space, having thicker cell walls, and a decrease in the amount of surface-associated fimbria (Figure 4E,F). Fimbria have been shown to play roles in both biofilm formation [60] and pathogenesis [61]; thus, the modifications seen in µG grown cell fimbria could alter both biofilm formation and pathogenesis. The increased cell wall thickness of the µG grown cells could impart resistance to osmotic shock or add protection against radiation exposure. These structural alterations seen in TEM are a direct stress response to the microgravity and elevated CO_2_ conditions, and they could possibly serve to benefit (or hinder) the yeast cell’s survivability in the spaceflight environment.

Additionally, the µG grown cells showed irregular intracellular organization with what appears to be large unknown membrane bound compartments/structures (Figure 4E). The function and content of these structures is still unclear, and it is currently unknown if these are vacuole or lysome-like vacuoles (Figure 4E) [62]. Previous reports have shown that some yeast can produce large numbers of compounds that are subsequently stored in vacuoles, while other yeast can utilize these large storage vesicles to degrade a large variety of substances [62,63]. Increased vacuole size could suggest a larger assortment and or quantity of nutrients being degraded by the Titan-like cells than by the standard gravity grown cells. Additionally, research has shown that portions of vacuoles can be passed to daughter cells [64]. Thus, it is possible that simulated microgravity induced compounds contained in the vacuoles may be passed to their progeny. Much further work is needed to better identify novel structures within the Titan-like cells, categorize the specific modifications that are occurring to organelles within the cells, determine how each of the cellular modifications alters the cells’ survivability in extreme environments (pH ranges, temperature ranges, osmotic shock, radiation exposure, etc.), and determine if these traits are passed on to their progeny.

*Cryptococcus* species have been shown to produce enlarged Titan-like cells to adapt to stress and contribute to *Cryptococcus* pathogenesis [20,65]. Both functions may also be of relevance for these ISS strains. The polyploidy nature of the yeast cells should be investigated as this might shed insights into the adaptation of these cells to microgravity conditions. Even though confocal microscopy measurements and increased fluorescence content in the microgravity-grown cells were documented, analysis of the nucleic acid reads of heterozygous regions of the genomes and a comparison of the ratios for the various alleles is needed to confirm the polyploidy nature of these cells. It is documented in *C. neoformans* that the enlarged polyploid Titan-like cells were able to generate progeny with aneuploidy. It was additionally found that the progeny of these Titan-like cells have both growth and survival advantages over typical cell progeny under stressful conditions [20]. Further gene characterizations are required to understand the molecular mechanisms associated with adaptation of novel IF6SW-B1^T^ yeast to space conditions [20].

While recent studies suggest that titanization is a rare phenomenon that occurs mostly in members of the *C. neoformans/C. gattii* species complex, the observation of Titan-like cells in ISS *Naganishia* species is novel. Understanding how microgravity influences this process is important to predict the emergence of virulence in *Naganishia* species and develop novel antifungal drugs. These antifungal drugs can be developed to inhibit the phenotypic transitions associated with fungal pathogenicity, especially since these strains were isolated across longitudinal sampling sessions of the ISS and appear persistent. Titanization and the increasing of cell size has been shown to aide *Cryptococcus* cells evade the immune system [21,26,65,66,67]. By growing from the normal range of ~1 to 3 μm in diameter to >10 μm, Titan cells cannot be readily ingested by phagocytic cells and, thus, persist in host tissue. Moreover, our data show that Titan-like cells are more likely to survive and produce offspring, when confronted by multiple environmental stress conditions, than typical cells. Additionally, their normally sized offspring maintained an advantage over regular cells in continued exposure to stress. Titan-like cell progeny’s increased survival and genomic diversity might promote rapid adaptation to new or high-stress space conditions; however, more research is needed to confirm this aspect.

### 3.3. Genomic Analyses of the Novel ISS Strain

Using BLAST, genes in ISS *Naganishia* species were compared to known genes that conferred resistance to cold, microgravity, and radiation (Table 1), as well as pigment biosynthesis. The reference genes were a compilation of NCBI queries and previously reported genes related to the spaceflight-related characteristics (Table 1). In addition to our four *N. tulchinskyi* strains, *N. vishniacii* (Antarctica strain), *Papiliotrema laurentii* (ISS isolates), and *Saccharomyces cerevisiae* (space flown strain) were included for comparison. A total of 24 genes aligned to one or more of the assemblies. Genes related to pigment synthesis and radioresistance were among the most prevalent, while genes associated with microgravity were mostly exclusive of *S. cerevisiae*.

Proteome prediction was performed with MAKER to determine biotechnologically important genetic elements in the four novel *Naganishia* strains [31]. Protein-coding genes were then assigned gene ontologies (GO) using InterProScan [68] (Figure 5) and passed through REVIGO [58] to summarize GO assignments and remove redundant terms (Appendix A). In total, 4894 protein-coding genes were predicted from the assembled draft genomes, which comprised 128 scaffolds (Appendix A). A total of 86, 47, and 89 top ontological hits were assigned, respectively, for Biological Process, Cellular Component, and Molecular Function (Appendix A), ranked by ontological uniqueness of term with respect to all other terms cataloged in UniProt. Top Biological Processes include cell morphogenesis, microtubule-based response, and response to UV light, reflecting a repertoire that was suited to respond to microgravity and radiation (Figure 5 and Appendix A). The extracellular region, outer membrane, and cell wall were top cellular component results, implying a set of genes associated with Titan-like cell plasticity. Finally, the top molecular function hits included cytoskeletal motor activity, microtubule motor activity, and nuclear export signal receptor activity. Changing these molecular mechanisms can alter the shape of cells, and in fungi, morphological changes in response to environmental stimuli have been reported and mostly attributed to pathogenesis [20,69]. The highest ontological hits revealed the presence of protein coding genes that might provide insights into the underlying origins of Titan-like cells in *N. tulchinskyi*. However, more studies are necessary to depict the mechanisms and role of titanization in microgravity.

The ability of *Naganishia* species to grow at low temperatures [29] and withstand high doses of UV radiation in extreme environments [1] has led to the ongoing microbial characterization of this genus; however, there are currently only a few genomes publicly available [70,71]. Unlike the *S. cerevisiae* genome, which has been extensively studied [72], the genomic characterization of the *Naganishia* clade is still in its infancy. The presence of reported genes related to cold resistance [39], radiation tolerance [47], and microgravity (Table 1) in ISS strains might indicate the potential of the novel *N. tulchinskyi* to better withstand the ISS or spaceflight-related conditions. However, only a few genes associated with microgravity have been identified [73,74]. It has been reported that the *RAD51*, *RAD54*, and *RDH54* genes seen in ISS strains also play critical roles in *C. neoformans* to repair gamma-radiation induced DNA damage [47].

Additionally, high carotenoid concentration has been correlated with an increased resistance of the cells against freeze damage, environmental stress, and radiation protection [75,76], all of which are notable features of the extremophile *Deinococcus radiodurans* [77,78]. Biomolecules responsible for the pigment biosynthesis, such as geranylgeranyl diphosphate synthase, cytochrome P450 reductase, farnesyl pyrophosphate synthase, isopentenyl-diphosphate delta isomerase, alpha, alpha-trehalase NTH2, and NTH1, putative trehalose-phosphate synthase Tps2 were present in all strains of *Naganishia* and *S. cerevisiae* (Table 1). It has been well documented that carotenoids can efficiently protect the skin from UV-light induced damages, scavenging singlet oxygen and peroxyl radicals [79]. The microbiological synthesis of carotenoids, significance, and biotechnological relevance were extensively reviewed [80]. The phytoene-beta carotene synthase (*pbs*) was present only in ISS strains but not found in *S. cerevisiae.* More research is needed here since biosynthesis of tetraterpenoids and *pbs* involvement in microorganisms is still poorly understood [81].

### 3.4. Taxonomy

Novel species: *Naganishia tulchinskyi,* N. K. Singh and K. Venkateswaran, sp. nov.

MycoBank number: MB838832.

**Etymology**: *Naganishia tulchinskyi* (tul.chin. s’ky.i. N.L. gen. n. *tulchinskyi* named after Igor Tulchinsky, an American Philanthropist supporting biomedical, computational, and space research).

**Diagnosis**: Based on the morphological characteristics, strains IF6SW-B1^T^, IF7SW-B1, IF1SW-F1, and IIF5SW-F1 are similar to *N. albida* group, but phylogenetically distant from all known *Naganishia* species. Hyphae and pseudohyphae were not present, sexual reproduction was not observed, budding cells were present and ballistoconidia were absent.

**Type strain**: IF6SW-B1^T.^

**Holotype**: USA: International Space Station surfaces, March 15, 2015. Nitin K. Singh and Kasthuri Venkateswaran, ex-type living culture (IF6SW-B1^T^) was preserved in a metabolically inactive state at the Northern Regional Research Laboratory (NRRL 64202) and at the Deutsche Sammlung von Mikroorganismen und Zellkulturen (DSM 113487).

**Description**: In PDA medium after 5 days at 25 °C, the cells are round, ovoid to ellipsoidal, 4.1 ± 0.97 μm and occur singly with polar budding. When grown under simulated microgravity at 30 °C, cells are ~3 times larger and formed Titan-like cells (Figure 2). After incubation at 25 °C, the streak culture on PDA agar is butyrous and cream colored, with a smooth surface and an entire margin. Growth occurs at 4, 25, and 30 °C and not at 35 °C on nutrient medium.

**Biochemical characteristics:** Differential biochemical characteristics of *N. tulchinskyi* with other closely related *Naganishia* species are shown in Table 3. *N. tulchinskyi* strains are able to assimilate D-Arabinose, D-Ribose, L-Sorbose, D-Mannitol, α-D-Glucose, L-Arabinose, Salicin, D-Trehalose, Maltose, D-Xylose, Sucrose, D-Cellobiose, Succinic Acid Mono-Methyl Ester, D-Xylose, and Palatinose as sole carbon source. *N. tulchinskyi* cells did not utilize Cellobiose, Arbutin, D-Galactose, Glycerol, Hexadecane, L-Malic acid, L-Rhamnose, Lactose, Mannitol, Melezitose, Methyl a-glucoside, Myo-inositol, Raffinose, Xylitol, Citrate, Melibiose, Meso erythritol, Glucono D-lactone, Inulin, and N-acetyl D-glucosamine as sole carbon source. *N. tulchinskyi* cells assimilate D-arabinose and can be differentiated from *N. friedmannii, N. saitoi, N. globosa.* Similarly, several carbon substrates are differentially utilized by *N. tulchinskyi* and are detailed in Table 3.

**Ecology/Substrate/Host**: Distinct *Naganishia* yeast colonies (*n* = 4) were isolated over two separate sample collection flights from three different locations. The type strain IF6SW-B1^T^ and two other strains (IF7SW-B1 and IF1SW-F1) were isolated during Flight 1 (March 2015), while the fourth strain (IIF5SW-F1) was isolated during Flight 2 (May 2015). Strain IF6SW-B1^T^ was isolated from a rack that contained crew clothes, crew preference items, office supplies, and ISS medical accessory kits (Location #6, Permanent Multipurpose Module Port 1). The second strain IF7SW-B1 was isolated from a rack being used for basic materials research in the microgravity environment of the ISS (Location #7, Overhead-3 panel surface of the Materials Science Research Rack 1). The third strain IF1SW-F1 was isolated from a viewing port in a small module devoted to the observation of operations outside the ISS, such as robotic activities, spacecraft approaches, and extravehicular activities (Location #1, Port panel of the Cupola). The fourth strain IIF5SW-F1 was the only strain in this study isolated during Flight 2, and it was cultivated from a storage rack made up of Nomex fabrics and used as a pantry to store batteries, printer cartridges, office supplies, etc. (Location #5, Overhead-4, Node 1, US module).

**Distribution**: This novel species has yet to be found in any samples from other Earth environments.

**Material examined**: The particulate matter associated with 1 m^2^ surface area of the ISS environmental surfaces was examined.

## 4. Conclusions

In summary, a novel yeast, *N. tulchinskyi*, was isolated from the ISS environmental surfaces that produced giant Titan-like cells under simulated microgravity conditions. Morphological characterization of *N. tulchinskyi* cells revealed that the cell wall is thicker under simulated microgravity and might help them withstand such unfavorable conditions, and these simulated microgravity-grown cells demonstrate a novel bud scar shedding trait. The phylogenetic analyses using the MLST approach showed that these are novel yeast. Further genomic characterization revealed genes responsible for cell morphogenesis and UVC resistance, reflecting a range of molecular mechanisms suited to respond to microgravity and radiation.

## Figures and Tables

**Figure 1 jof-08-00165-f001:**
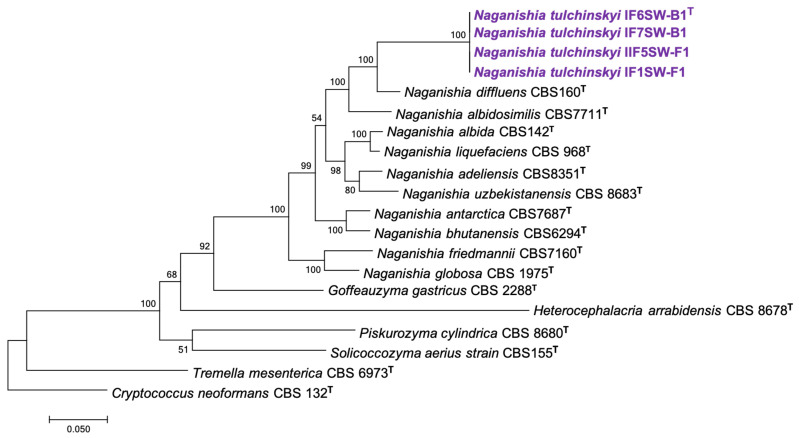
Maximum Likelihood phylogenetic tree, based on seven gene sequences (ITS, LSU, SSU, *CYTB*, *RPB1*, *RPB2*, *TEF1*), extracted from WGS of the respective type strains (superscript T) or from the available genomes, concatenated manually, showing the phylogenetic relationship of *N. tulchinskyi* sp. nov. with closely related members of *Tremellomycetes*. The gene fragment lengths in base pairs (bp) are given in parenthesis: ITS (620 bp); LSU (612 bp); SSU (517 bp); *CYTB* (339 bp); *RPB1* (317 bp); *RPB2* (1282 bp); *TEF1* (1084 bp). Bootstrap values from 1000 replications are shown at branch points. Bar = 0.05 substitution per site.

**Figure 2 jof-08-00165-f002:**
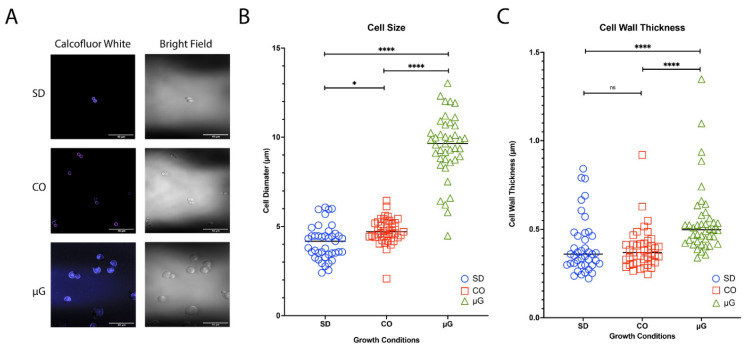
Confocal microscope-based analyses of *N. tulchinskyi* cells when grown under standard Earth gravity (SD), normal gravity with 5% CO_2_ (CO), and simulated microgravity with 5% CO_2_ (µG). Representative images of Calcofluor White cell wall-stained cells adjacent to bright field images are shown in (**A**). The µG cells appear to be larger and misshapen when compared to the typical yeast morphology demonstrated by SD and CO. Measurements for cell diameter were taken using the of Calcofluor White confocal images (**B**). Though smaller than µG cells, SD cells were statically smaller in cell size than CO cells (*p* value = 0.0440), while µG cells were statistically significantly larger than both SD and CO cells (*p* value ≤ 0.0001). Cell wall thickness was also measured using confocal microscopy (**C**). Cell wall thickness of SD and CO were not statistically different in size (*p* value ≥ 0.9999); however, µG cells had statistically significantly larger cell walls than both SD and CO cells (*p* value ≤ 0.0001). The scale bars for both confocal and bright field microscopic images is 40 µm. Statistical significance is reported as *p* value > 0.05 (ns), ≤0.05 (*), ≤0.01 (**), ≤0.001 (***), ≤0.0001 (****). *p* values of ≤0.01 (**) and ≤0.001 (***) were not noticed.

**Figure 3 jof-08-00165-f003:**
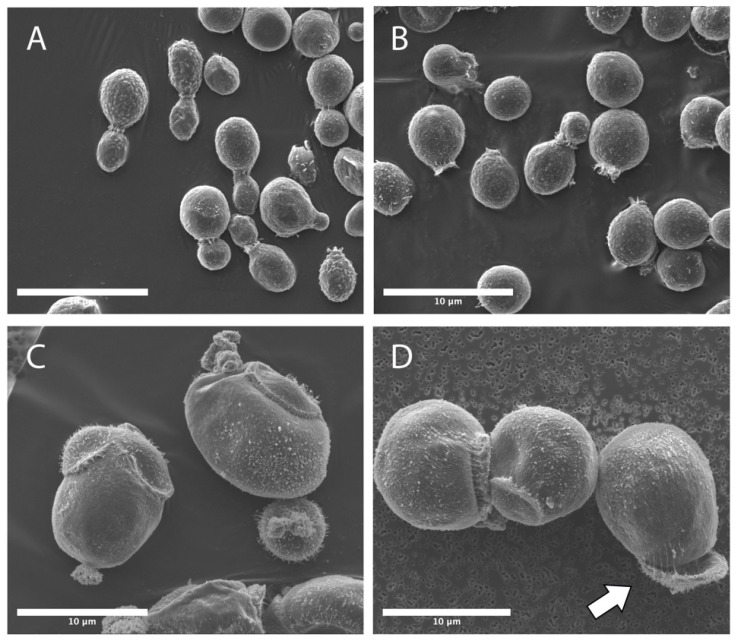
Analyses of scanning electron microscope (SEM) micrographs of SD, CO, and µG grown *N. tulchinskyi* cells demonstrated differences in conidia size, morphology, and daughter cell budding. SD (**A**) and CO (**B**) grown cells demonstrated similar standard yeast morphotypes, while µG (**C**,**D**) grown cells are larger, nonuniform in appearance, and exhibit random budding scars. In some instances, the µG budding scars have raised centers, while others appear to be concave. Additionally, some budding scars appear to be shedding or peeling from the surface of the cells. This phenomenon is highlighted with the white arrow (**D**). Representative SEM micrographs are given, and scale bar for all micrographs is 10 µm.

**Figure 4 jof-08-00165-f004:**
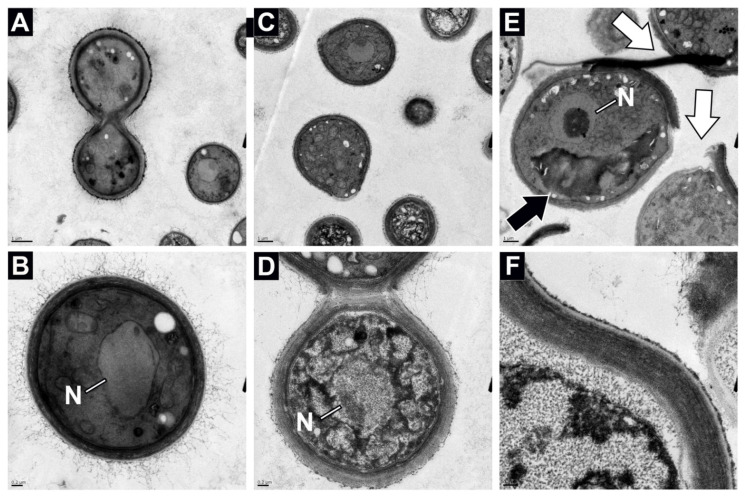
Analyses of transmission electron microscope (TEM) micrographs of SD (**A**,**B**), CO (**C**,**D**), and µG (**E**,**F**) grown *N. tulchinskyi* cells demonstrated differences in cell size, cell wall thickness, and internal organelle structures. SD and CO demonstrate similar size cells and similar intracellular morphology, while µG grown cells appear to have a different internal organizational structure of organelles. At higher magnifications (lower panels), the cell wall thickness can be seen to be thicker for µG grown cells than for SD or CO grown cells. The cells also appear to be having lower abundance of fine scale fimbriae on the surface of µG cells, as compared to SD and CO grown cells. Along with their large size, the µG grown cells are distinguished by the shedding and peeling bud scars from their surfaces, denoted by white arrows (**C**). Additionally, µG grown cells have the presence of large unknown organelles/structures, as denoted by the black arrow. Representative TEM micrographs are given. The scale bar for (**A**,**C**,**E**) is 1 µm, and the scale bare for (**B**,**D**,**F**) is 0.2 µm.

**Figure 5 jof-08-00165-f005:**
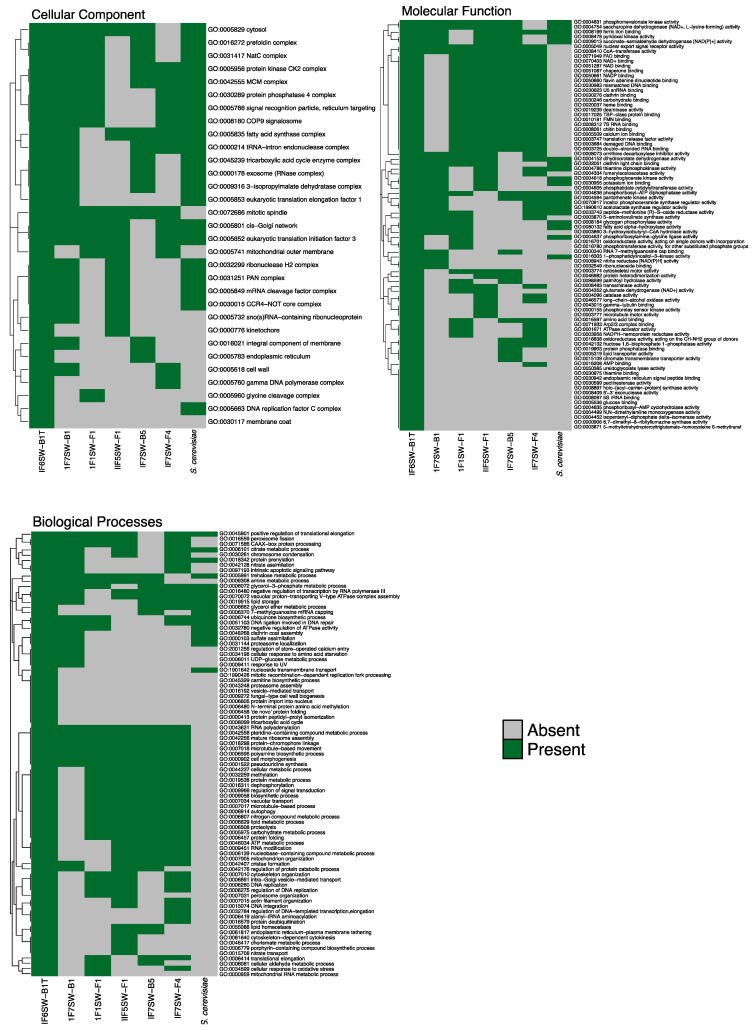
Enriched GOSlim terms for biological process, cellular components, and molecular function in *N. tulchinskyi* genome. The Gene Ontology terms were summarized by purging redundant GO terms that were deduced by orthologous searches coupled with semantic similarity-based scatterplots using the REVIGO platform [58].

**Table 3 jof-08-00165-t003:** Physiological characteristics of *Naganishia tulchinskyi* and closely related species.

Carbon Source		*Naganishia tulchinskyi* (This Study)				*N. vaughanmartiniae* ^a^	*N. onofrii* ^a^	*N. saitoi* ^b^	*N. friedmannii* ^b^	*N. qatarensis* ^c^	*N. cerealis* ^d^	*N. randhawae* ^e^	*N. globosa* ^b^
		IF6SW-B1	IF7SW-B1	IF1SW-F1	IIF5SW-F1								
**Growth at:**													
	4 °C	+	+	+	+	+	+	NT	+	NT	NT	NT	NT
	25 °C	+	+	+	+	+	+	+	-	+	+	+	+
	30 °C	+	+	+	+	-	-	+	-	+	+	+	+
	35 °C	-	-	-	-	-	-	-	-	+	-	-	-
**Differential characteristics among various *Naganishia* species**													
	D-Arabinose	+	+	+	+	+	+	-	-	+	+	w	-
	D-Ribose	+	+	+	+	+	v	v	-	+	v	-	-
	L-Sorbose	+	+	+	+	v	v	-	-	+	+	-	-
	D-Mannitol	+	+	+	-	NT	NT	NT	-	+	+	+	+
	α-D-Glucose	+	+	+	+	+	+	+	+	-	NT	NT	NT
	L-Arabinose	+	+	+	+	+	+	+	+	+	w	w	-
	Salicin	-	+	+	+	+	+	+	+	-	NT	NT	NT
	D-Trehalose	+	+	-	+	+	+	+	+	-	NT	NT	NT
	Maltose	+	+	+	-	+	+	+	+	-	NT	NT	NT
	D-Xylose	+	+	-	+	+	+	+	+	-	NT	NT	NT
	Sucrose	+	+	-	+	+	+	+	v	-	NT	NT	NT
	Cellobiose	-	-	-	-	+	+	+	+	-	NT	NT	NT
	Arbutin	-	-	-	-	+	+	NT	NT	-	NT	NT	NT
	D-Galactose	-	-	-	-	+	+	-/w	-	+	+	+	w
	Glycerol	-	-	-	-	v	+	-/s	-	+	-	-	-
	Hexadecane	-	-	-	-	-	-	NT	NT	NT	NT	NT	NT
	L-Malic acid	-	-	-	-	-	-	+	v	NT	NT	NT	NT
	L-Rhamnose	-	-	-	-	+	+	+	-	+	+	+	+
	Lactose	-	-	-	-	v	+	v	-	+	+	+	v
	Mannitol	-	-	-	-	+	+	+	-	+	-	+	-
	Melezitose	-	-	-	-	+	+	+	+	-	NT	NT	NT
	Methyl a-glucoside	-	-	-	-	+	+	+	v	-	NT	NT	NT
	Myo-inositol	-	-	-	-	v	+	+	-	+	+	+	+
	Raffinose	-	-	-	-	-/w	+	-/w	-	+	w	+	w
	Xylitol	-	-	-	-	+	+	-	-	+	+	+	-
	Citrate	-	-	-	-	-	-	+	v	-	-	-	+
	Melibiose	-	-	-	-	-	-	-	-	+	-	-	-
	Meso erythritol	-	-	-	-	-	-	-	-	w	-	-	-
	Glucono D-lactone	-	-	-	-	-	v	NT	NT	+	+	-	NT
	*N*-acetyl D-glucosamine	-	-	-	-	-	-	NT	NT	-	NT	NT	NT
**Common characteristics among *N. tulchinskyi* strains**													
	D-Cellobiose	+	+	+	+								
	Succinic Acid Mono-Methyl Ester plus D-Xylose	+	+	+	+								
	Palatinose	+	+	+	+								
	Inulin	-	-	-	-								
**Differential characteristics among *N. tulchinskyi* strains**													
	Gentiobiose	+	+	+	-								
	D-Mannitol	+	+	+	-								
	Maltose	+	+	+	-								
	Acetic Acid	+	+	-	+								
	Maltotriose	+	+	-	+								
	Turanose	+	+	-	+								
	D- Glucuronic Acid plus D-Xylose	+	+	-	+								
	1,2- Propanediol plus D-Xylose	+	+	-	+								
	D-Trehalose	+	+	-	+								
	D-Xylose	+	+	-	+								
	Sucrose	+	+	-	+								
	D-Melezitose	-	+	+	+								
	Dextrin	-	+	+	+								
	Propionic Acid	+	+	-	-								
	D-Raffinose	-	-	+	-								
	D-Sorbitol	-	-	+	-								

^a^ Data from Turchetti et al. 2015 (11). ^b^ Data obtained from Fonseca et al. 2011 (13). ^c^ Data obtained from Fotedar et al. 2018 (1). ^d^ Data obtained from Passoth et al. 2009 (16). ^e^ Data obtained from Khan et al. 2010 (14). Abbreviation: “+” positive; “-” negative; “NT” not tested; “w” weak reaction; “v” variable reaction.

## Data Availability

The WGS and raw data have been deposited in GenBank under the BioProject accession number PRJNA623412 [6]. The genome sequences has also been deposited in the NASA GeneLab system (GLDS-290; https://genelab-data.ndc.nasa.gov/genelab/accession/GLDS-290; accessed on 5 December 2021).

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
