# Peer review of "Genomic Characterization of the Titan-like Cell Producing Naganishia tulchinskyi, the First Novel Eukaryote Isolated from the International Space Station"

_jof, 2022, doi:10.3390/jof8020165_

Round 1

Reviewer 1 Report

Two main questions: 1) You need to indicate whether the reference sequences of all species are derived from type specimens. For taxonomic paper, all sequences for phylogenetic analyses were derived from type materials; 2) Since you believe ITS, and LSU genes might not be usefulto differentiate Naganishia species, why your phylogenetic analyses were also based on seven gene loci.

Author Response

1) You need to indicate whether the reference sequences of all species are derived from type specimens. For taxonomic paper, all sequences for phylogenetic analyses were derived from type materials;

  • Ans: All reference sequences were deduced from the type specimens and are denoted with superscript “T” after the strain number. This information was added in Figure 1 of the new manuscript.

2) Since you believe ITS, and LSU genes might not be useful to differentiate Naganishia species, why your phylogenetic analyses were also based on seven gene loci.

  • Ans: The precedence set for MLST scheme developed for Tremellomycetes by Liu et al. 2015 suggest that we should use seven-genes loci. Authors want to follow the established molecular approach when determining novel basidiomycetes species.
  • The taxonomic threshold predicted to discriminate basidiomycetous yeast species is 99.51% in the LSU region and 99.21% in the ITS region [Fotedar et al., 2018]. This is too high of similarity between the new species and the already established species. Hence Liu et al. 2015 suggested 7 genes loci to differentiate basidiomycetous yeast species.

Reviewer 2 Report

Authors improved the content significantly. My only concern is the quality of ITS sequences. The deposited sequences of IF1SW-F1 (KY218664.1) and IF6SW-B1 (KY218715.1) still differ notably from the ITS regions found in contigs. The ITS tree (Suppl Fig 1) separates only IF6SW-B1 strain, while Fig 1 separates both (IF1SW-F1 and IF6SW-B1).

The contradiction can be resolved with a simple ITS1-ITS4 sequencing on both strands. After an appropriate reaction the carefully edited sequence should be used in the analysis, and the flat files (KY218664.1 and KY218715.1) should be corrected (sequences and species name) accordingly.

The collection acronym and accession number of the holotype must be specified. If the holotype is a living culture, it must be stated (if true) that it is preserved in a metabolically inactive state. The absence of this information renders the description of the novel species to be invalid.

Author Response

Authors improved the content significantly. My only concern is the quality of ITS sequences. The deposited sequences of IF1SW-F1 (KY218664.1) and IF6SW-B1 (KY218715.1) still differ notably from the ITS regions found in contigs. The ITS tree (Suppl Fig 1) separates only IF6SW-B1 strain, while Fig 1 separates both (IF1SW-F1 and IF6SW-B1).

The contradiction can be resolved with a simple ITS1-ITS4 sequencing on both strands. After an appropriate reaction the carefully edited sequence should be used in the analysis, and the flat files (KY218664.1 and KY218715.1) should be corrected (sequences and species name) accordingly.

  • Ans: As per reviewer’s suggestion we have re-sequenced the ITS region and used the new ITS sequences to generate MLST tree (Figure 1) and ITS tree (Supplemental Figure S1). As predicted by the reviewer, there are Sanger sequencing errors and are rectified in the modified manuscript. The ITS-based tree now shows only one tulchinskyi clade.

The collection acronym and accession number of the holotype must be specified. If the holotype is a living culture, it must be stated (if true) that it is preserved in a metabolically inactive state. The absence of this information renders the description of the novel species to be invalid.

  • Ans: We have provided this information in Lines 475 to 476 and reproduced below.
    • ex-type living culture (IF6SW-B1T) was preserved in a metabolically inactive state at the Northern Regional Research Laboratory (NRRL 64202) and at the Deutsche Sammlung von Mikroorganismen und Zellkulturen (DSM 113487).

Reviewer 3 Report

Manuscript ID: jof-1548677

Title: Genomic characterization of the Titan-like cell producing Naganishia tulchinskyi, the first novel eukaryote isolated from the International Space Station

In this article, the authors describe Genomic characterization of the Titan-like cell producing Naganishia tulchinskyi, the first novel eukaryote isolated from the International Space Station. I think this article is very interesting. The manuscript is generally well written and structured. However, some revisions are required to improve the quality of the content and writing.

The following comments are for the authors to consider:

L57: Delete repeat word

L90-94: How long were the samples transported to Earth and stored until subjected to yeast isolation?

L100-103: Please indicate stain code of the six strains in this sentence.

L120-121: How many nucleotide positions were taken into account for the analysis and how many were in the alignment?

L123: Sexual reproduction and mycelial formation must be tested in this section. In addition, there are other tests (e.g. nitrogen source assimilation, fermentation of carbohydrates, antibiotic resistance and growth tests) for yeast description that could improve the quality of this study.
Suggest to follow:

Kurtzman CP, Fell JW, Boekhout T, Robert V. Methods for isolation, phenotypic characterization and maintenance of yeasts. In: Kurtzman CP, Fell JW, Boekhout T (editors). The Yeasts, a Taxonomic Study. 5th edn. Amsterdam: Elsevier; 2011. pp. 87–110.

L125-129: The authors should provide the composition of culture media

L131: the second set

L141-148: In this study, the biochemical and physiological characterizations were tested by using test kit (BioLog Yeast plates) and incubated at 25ËšC for 7 days. I think it is not appropriate for use in describing yeast. In general, biochemical and physiological characterizations take about 28 days on average. I strongly recommend you to follow:

Kurtzman CP, Fell JW, Boekhout T, Robert V. Methods for isolation, phenotypic characterization and maintenance of yeasts. In: Kurtzman CP, Fell JW, Boekhout T (editors). The Yeasts, a Taxonomic Study. 5th edn. Amsterdam: Elsevier; 2011. pp. 87–110.

L149: Please indicate the age of yeast cells and the cultured medium

L210: It would be nice to indicate species of the two closely related species in this sentence.

L255: The sequence comparison (e.g. number of nucleotide divergence in the D1/D2 domain of LSU rRNA and ITS regions) between the proposed new species and its neighbor species need to be explained in this section.

L256: Suggest to provide the current name of Cryptococcus diffluens. It's possible to put it in parentheses

L257: The authors should explain about these two independent flight collection studies and the four different ISS locations in more detail. Please clarify where each of these strains IF6SW-B1T, IF7SW-B1, IF1SW-F1, and IIF5SW-F1 isolated from. It's possible to put it in the Table.

L282-285: The authors stated that “The ITS and LSU-based phylogenetic trees (Supplemental Figures S1 and S2) displayed two clades; two strains showed N. albida as the closest neighbor and the other two strains were affiliated to N. diffluens CBS 160, indicating ITS, and LSU genes might not be useful to differentiate Naganishia species. When tested individually, the phylogenetic analyses of the other five genes (SSU, CYTB, RPB1, RPB2, and TEF1) formed a single group for all four ISS strains.”. In my personal opinion, these sentences are inconsistent with the Supplemental Figures. The ITS tree (Suppl Figure 1) showed that the three strains (IF1SW-F1, IIF5SW-F1 and IF7SW-B1) were placed at the same position of the three recognized species, Naganishia diffluens, N. liquefaciens and N. albidosimilis, while the strain IF6SW-B1 was speared from the others. In the LSU tree (Suppl Figure 2), these four strains were placed at the same position that phylogenetically distinguished from Naganishia albida, N. adeliensis, N. vishniacii and N.albidosimilis with low bootstrap support. The SSU tree (Suppl Figure 3) demonstrated that these four strains were not separated from the recognized species. Moreover, the Suppl Figures 1-3 were not clear that what species is the closely related species for the four strains. Thus, I suggest to revise these sentences.

L474-476: Nomenclatural rules applicable to yeasts are found in the ‘International Code of Nomenclature for algae, fungi, and plants’ (ICNafp, or the Code) (Turland et al. 2018).

According to the ICNafp, the Code recommends the deposition of living cultures prepared from the holotype in at least two culture collections (Recommendation 8B1). In this study, the type strain was deposited at only one culture collection (NRRL). Thus, the authors must deposit the type strain at least 1 additional culture collection in a difference country.

In addition, when a culture is designated as a type, the status of the culture should be indicated, including the phrase “permanently preserved in a metabolically inactive state“ or an equivalent. Thus, I suggest to replace “ ex-type living culture (IF6SW-B1T) was deposited at the Northern Regional Research Laboratory (NRRL)” with “ ex-type living culture (IF6SW-B1T) was preserved in a metabolically inactive state at the Northern Regional Research Laboratory (NRRL)”.

Turland NJ, Wiersema JH, Barrie FR, Greuter W, Hawksworth DL, Herendeen PS, Knapp S, Kusber W-H, Li D-Z, Marhold K, May TW, McNeill J, Monro AM, Prado J, Price MJ, Smith GF (2018) International Code of Nomenclature for algae, fungi, and plants (Shenzhen Code) adopted by the Nineteenth International Botanical Congress Shenzhen, China, July 2017. Regnum Vegetabile 159. Koeltz Botanical Books, Glashütten. https:// doi. org/ 10. 12705/ Code. 2018

L477: The authors mentioned “In PDA medium after 5 days at 25ËšC, the conidia are globose, ovoid to ellipsoidal...”. This yeast species produced conidia?  I am not sure what did you mean, vegetative cells or asexual spore? Please clarify

Figure 1.: The authors should indicate the proposed name (Naganishia tulchinskyi) in the tree.

Table 3: What is “?/w” mean?

Author Response

In this article, the authors describe Genomic characterization of the Titan-like cell producing Naganishia tulchinskyi, the first novel eukaryote isolated from the International Space Station. I think this article is very interesting. The manuscript is generally well written and structured. However, some revisions are required to improve the quality of the content and writing.

The following comments are for the authors to consider:

L57: Delete repeat word

  • Ans: deleted.

L90-94: How long were the samples transported to Earth and stored until subjected to yeast isolation?

  • Ans: Information about the sample collection/processing/isolation of fungi were published and appropriately cited in this manuscript (Ref 5). The following sentences were added to clarify the information.
    • Samples were collected just prior to departure (within 5 days) and stored at 4ËšC in the ISS, transported in cold packs (~4ËšC) from ISS to the Earth, hand delivered at Long Beach, California after splash down, which were then carried in ice packs (~4ËšC) during ground transportation, before storing at JPL at 4ËšC for further processing. Total number of days for sample storage time from sample collection to processing were ~7 days.

L100-103: Please indicate stain code of the six strains in this sentence.

  • Assume reviewer is asking the strain codes (or strain numbers) for four strains characterized during this study and they are given in Data availability section and also in all the phylogenetic tree. See Lines 263 to 264. Reproduced below as well.
    • Naganishia IF6SW-B1 (KY218715.1); Naganishia IF1SW-F1 (KY218664.1); Naganishia IF7SW-B1 (KY218717.1); Naganishia IIF5SW-F1 (KY218695.1)

L120-121: How many nucleotide positions were taken into account for the analysis and how many were in the alignment?

  • Ans: The gene fragment lengths in base pairs are given in Figure 1.
    • ITS (30 strains for alignment) – 620 bases
    • LSU (26 strains for alignment) – 612 bases
    • SSU (24 strains for alignment) – 517 bases
    • CYTB (22 strains for alignment) – 339 bases
    • RPB1 (23 strains for alignment) – 317 bases
    • RPB2 (21 strains for alignment) – 1282 bases
    • TEF1 (19 strains for alignment) – 1084 bases
    • Number of species aligned are given in each and every figure. For the MLST analyses there were 14 Naganishia species, 3 closely related genera, and 1 outgroup Cryptococcus neformans were used for alignment.

L123: Sexual reproduction and mycelial formation must be tested in this section.

  • Ans: The following sentence added in the section 3.2 Taxonomy
    • Hyphae and pseudohyphae were not present, sexual reproduction was not observed, budding cells were present and ballistoconidia were

L123: In addition, there are other tests (e.g. nitro-gen source assimilation, fermentation of carbohydrates, antibiotic resistance and growth tests) for yeast description that could improve the quality of this study.

  • Ans: BioLog test (n=94) consists of testing assimilation of various nitrogen sources, fermentation of several carbohydrate substrates. We did not test antibiotic resistant profiles. Growth at various temperatures was tested.

L125-129: The authors should provide the composition of culture media

  • Ans: YPD agar medium composition is provided in the modified manuscript.

L131: the second set

  • Ans: Corrected.

L141-148: In this study, the biochemical and physiological characterizations were tested by using test kit (BioLog Yeast plates) and incubated at 25ËšC for 7 days. I think it is not appropriate for use in describing yeast. In general, biochemical and physiological characterizations take about 28 days on average. I strongly recommend you to follow:

Kurtzman CP, Fell JW, Boekhout T, Robert V. Methods for isolation, phenotypic characterization and maintenance of yeasts. In: Kurtzman CP, Fell JW, Boekhout T (editors). The Yeasts, a Taxonomic Study. 5th edn. Amsterdam: Elsevier; 2011. pp. 87–110.

  • Ans: Authors understand the importance of doing traditional way using test tubes for characterizing biochemical tests. However, recent advances of using phenotypic array (94 tests in BioLog) specifically for identifying yeast becoming common. Hence, we have used the BioLog technology. As seen by the reviewer, except few, the molecular technologies using sequencing approaches are more confirmative in identifying novel microorganisms including yeasts and fungi. The biochemical tests are carried out to augment the conclusion derived from the molecular techniques. Even though manufacturer recommends to score results after 72 hours, we have incubated BioLog plates for more than 30 days. The BioLog test results obtained after 72 hours remain the same after incubation day 7, 14, and/or at 30.
  • Use of BioLog for identifying yeast are not new and was established since 1997. Some examples are given below.
    • Praphailong, W.; Van Gestel, M.; Fleet, G.H.; Heard, G.M. Evaluation of the Biolog system for the identification of food and beverage yeasts. Appl. Microbiol. 1997, 24, 455-459, doi:10.1046/j.1472-765x.1997.00057.x.
    • Pincus, D.H.; Orenga, S.; Chatellier, S. Yeast identification — past, present, and future methods. Mycol. 2007, 45, 97-121, doi:10.1080/13693780601059936.
    • Sha, S.P.; Suryavanshi, M.V.; Jani, K.; Sharma, A.; Shouche, Y.; Tamang, J.P. Diversity of Yeasts and Molds by Culture-Dependent and Culture-Independent Methods for Mycobiome Surveillance of Traditionally Prepared Dried Starters for the Production of Indian Alcoholic Beverages. Frontiers in microbiology 2018, 9, 2237-2237, doi:10.3389/fmicb.2018.02237.

L149: Please indicate the age of yeast cells and the cultured medium

  • Ans: Samples were grown in potato dextrose broth for 7 days. This is added in the modified manuscript

L210: It would be nice to indicate species of the two closely related species in this sentence.

  • Ans: diffluens and N. albidosimilis were the two closely related species based on MLST but that will be results and given in Figure 1. Since this section is Materials and Methods, we did not mention it here. However, if reviewer need this information in Methods, we can add them.

L255: The sequence comparison (e.g. number of nucleotide divergence in the D1/D2 domain of LSU rRNA and ITS regions) between the proposed new species and its neighbor species need to be explained in this section.

  • Ans: The difference between tulchinskyi type strain with the type strains of nearest species are given. The relevant sentences added in the modified manuscript are reproduced below.
  • The ITS tree (Supplemental Figure 1) showed that the ISS strains were placed at the same position of the three recognized species, Naganishia diffluens, N. liquefaciens and  albidosimilis. ITS sequences of the four ISS isolates showed 100% similarity to the sequence of N. albidosimilis CBS 7711T (AF145325), N. liquefaciens CBS 968T (AF444345), and N. diffluens CBS 160T (AF145330). However, ITS sequences of N. vishniacii CBS 7110T shared 97.84% similarity (13 bp difference to GenBank accession no. AF145320.), N. uzbekistanensis CBS 8683T had 97.55% similarity (14 bp difference to GenBank accession no. AF444339), and Naganishia adeliensis CBS 8351T exhibited 97.34% similarity (14 bp difference to GenBank accession no. AF145328) with ISS isolates (Supplemental Figure S1). In the LSU tree (Supplemental Figure 2), the four ISS strains were placed at the same position that phylogenetically distinguished from Naganishia albida, N. adeliensis, N. vishniacii andN. albidosimilis with low bootstrap support. Comparing the LSU sequences of these ISS strains with closely related species showed a 97.71% similarity to N. albida (type strain, CBS 142; 14 different base pairs; GenBank accession no. AF075474), 96.41% similarity to N. albidosimilis (type strain, CBS 1926; 22 different base pairs; GenBank accession no. AF137601), 96.25% similarly to N. liquefaciens (type strain, CBS 968; 23 different base pairs; GenBank accession no. AF181515) and 95.92% similarity to N. diffluens (type strain, CBS 160; 25 different base pairs; GenBank accession no. AF075502) (Supplemental Figure S2). The taxonomic threshold predicted to discriminate basidiomycetous yeast species is 99.51% in the LSU region and 99.21% in the ITS region [1]. Hence, the sequence data and phylogenetic analysis of the LSU regions of the ISS isolates confirm they belong to a new species in the genus Naganishia (order Filobasidiales).

L256: Suggest to provide the current name of Cryptococcus diffluens. It's possible to put it in parentheses

  • Ans: Added

L257: The authors should explain about these two independent flight collection studies and the four different ISS locations in more detail. Please clarify where each of these strains IF6SW-B1T, IF7SW-B1, IF1SW-F1, and IIF5SW-F1 isolated from. It's possible to put it in the Table.

  • Ans: The following sentences were added in the modified manuscript. This may go to methods, if the reviewer does not want to place in the Results section.
    • The sampling of ISS surfaces performed for this study took place within the US on-orbit segments. Sampling campaign (denoted as I) was carried out on March 4th 2015 and second one (denoted as II) was conducted on May 15th Samples collected during this study were: Node 3 (Locations #1, #2 and #3), Node 1 (Locations #4 and #5), Permanent Multipurpose Module (Location #6), U.S. Laboratory (Location #7), and Node 2 (Locations #8 and control). A detailed description of the various locations sampled was published elsewhere [42]. Strains IF6SW-B1, IF7SW-B1, and IF1SW-F1, were isolated during first sampling from Locations 6, 7, and 1 respectively. Strain IIF5SW-F1 was isolated from second sampling at Location 5.

L282-285: The authors stated that “The ITS and LSU-based phylogenetic trees (Supplemental Figures S1 and S2) displayed two clades; two strains showed N. albida as the closest neighbor and the other two strains were affiliated to N. diffluens CBS 160, indicating ITS, and LSU genes might not be useful to differentiate Naganishia species. When tested individually, the phylogenetic analyses of the other five genes (SSU, CYTB, RPB1RPB2, and TEF1) formed a single group for all four ISS strains.”. In my personal opinion, these sentences are inconsistent with the Supplemental Figures. The ITS tree (Suppl Figure 1) showed that the three strains (IF1SW-F1, IIF5SW-F1 and IF7SW-B1) were placed at the same position of the three recognized species, Naganishia diffluens, N. liquefaciens and N. albidosimilis, while the strain IF6SW-B1 was speared from the others. In the LSU tree (Suppl Figure 2), these four strains were placed at the same position that phylogenetically distinguished from Naganishia albida, N. adeliensis, N. vishniacii and N.albidosimilis with low bootstrap support. The SSU tree (Suppl Figure 3) demonstrated that these four strains were not separated from the recognized species. Moreover, the Suppl Figures 1-3 were not clear that what species is the closely related species for the four strains. Thus, I suggest to revise these sentences.

  • Ans: After re-sequencing all four strains (as per one of the reviewers), we were able to fill the sequence gaps and the new ITS tree is showing only one clade and hence the manuscript is revised.
  • As suggested by the reviewer, we modified these sentences and are reproduced below.

The ITS tree (Supplemental Figure 1) showed that the ISS strains were placed at the same position of the three recognized species, Naganishia diffluens, N. liquefaciens and N. albidosimilis. ITS sequences of the four ISS isolates showed 100% similarity to the sequence of N. albidosimilis CBS 7711T (AF145325), N. liquefaciens CBS 968T (AF444345), and N. diffluens CBS 160T (AF145330). However, ITS sequences of N. vishniacii CBS 7110T shared 97.84% similarity (13 bp difference to GenBank accession no. AF145320.), N. uzbekistanensis CBS 8683T had 97.55% similarity (14 bp difference to GenBank accession no. AF444339), and Naganishia adeliensis CBS 8351T exhibited 97.34% similarity (14 bp difference to GenBank accession no. AF145328) with ISS isolates (Supplemental Figure S1). In the LSU tree (Supplemental Figure 2), the four ISS strains were placed at the same position that phylogenetically distinguished from Naganishia albida, N. adeliensis, N. vishniacii and N. albidosimilis with low bootstrap support. Comparing the LSU sequences of these ISS strains with closely related species showed a 97.71% similarity to N. albida (type strain, CBS 142; 14 different base pairs; GenBank accession no. AF075474), 96.41% similarity to N. albidosimilis (type strain, CBS 1926; 22 different base pairs; GenBank accession no. AF137601), 96.25% similarly to N. liquefaciens (type strain, CBS 968; 23 different base pairs; GenBank accession no. AF181515) and 95.92% similarity to N. diffluens (type strain, CBS 160; 25 different base pairs; GenBank accession no. AF075502) (Supplemental Figure S2). The taxonomic threshold predicted to discriminate basidiomycetous yeast species is 99.51% in the LSU region and 99.21% in the ITS region [1]. Hence, the sequence data and phylogenetic analysis of the LSU regions of the ISS isolates confirm they belong to a new species in the genus Naganishia (order Filobasidiales). The SSU tree (Supplemental Figure 3) demonstrated that these four strains were not separated from any of the recognized species. The other phylogenetic trees based on CYTB, RPB1, RPB2, and TEF1 grouped the ISS strains together, and the closest neighbor was identified as N. diffluens.

L474-476: Nomenclatural rules applicable to yeasts are found in the ‘International Code of Nomenclature for algae, fungi, and plants’ (ICNafp, or the Code) (Turland et al. 2018).

According to the ICNafp, the Code recommends the deposition of living cultures prepared from the holotype in at least two culture collections (Recommendation 8B1). In this study, the type strain was deposited at only one culture collection (NRRL). Thus, the authors must deposit the type strain at least 1 additional culture collection in a difference country.

In addition, when a culture is designated as a type, the status of the culture should be indicated, including the phrase “permanently preserved in a metabolically inactive state“ or an equivalent. Thus, I suggest to replace “ ex-type living culture (IF6SW-B1T) was deposited at the Northern Regional Research Laboratory (NRRL)” with “ ex-type living culture (IF6SW-B1T) was preserved in a metabolically inactive state at the Northern Regional Research Laboratory (NRRL)”.

Turland NJ, Wiersema JH, Barrie FR, Greuter W, Hawksworth DL, Herendeen PS, Knapp S, Kusber W-H, Li D-Z, Marhold K, May TW, McNeill J, Monro AM, Prado J, Price MJ, Smith GF (2018) International Code of Nomenclature for algae, fungi, and plants (Shenzhen Code) adopted by the Nineteenth International Botanical Congress Shenzhen, China, July 2017. Regnum Vegetabile 159. Koeltz Botanical Books, Glashütten. https:// doi. org/ 10. 12705/ Code. 2018

  • Ans: We deposited the type strain (IF6SW-B1T) in two culture collections (NRRL and DSMZ). We added the following sentence in the modified manuscript.
    • ex-type living culture (IF6SW-B1T) was preserved in a metabolically inactive state at the Northern Regional Research Laboratory (NRRL 64202) and at the Deutsche Sammlung von Mikroorganismen und Zellkulturen (DSM 113487).

L477: The authors mentioned “In PDA medium after 5 days at 25ËšC, the conidia are globose, ovoid to ellipsoidal...”. This yeast species produced conidia?  I am not sure what did you mean, vegetative cells or asexual spore? Please clarify

  • This is inadvertently placed. These are not forming conidia. The modified description placed in the manuscript are reproduced below.
    • In PDA medium after 5 days at 25ËšC, the cells are round, ovoid to ellipsoidal, 4.1 ± 0.97 μm and occur singly with polar budding.

Figure 1.: The authors should indicate the proposed name (Naganishia tulchinskyi) in the tree.

  • Added in Figure 1.

Table 3: What is “?/w” mean?

  • It should be “-/w” and is modified in the revised manuscript.

Round 2

Reviewer 3 Report

The comments have been addressed and the manuscript was revised clearly. So, this revised manuscript can be accepted for publication.

This manuscript is a resubmission of an earlier submission. The following is a list of the peer review reports and author responses from that submission.

Round 1

Author Response

Since reviewer's comments were in the word file, we corrected then and there in the revised manuscript and the same is attached.

Reviewer 2 Report

The authors characterize four strains sampled from the International Space Station and propose that these are representatives of a new undescribed species of the genus Naganishia. Interestingly the authors show that these cells adopt a titan-like phenotype when grown under simulated microgravity combined with 5% CO2 conditions. This is a phenotype that has been previously associated to pathogenicity in Cryptococcus yeasts.
The carefully annotation of these new genomes and the detailed characterization of the titan-like phenotype in the cells morphology is novel and interesting. Yet, the authors have not provided compelling information that these strains are in fact from a new species. (1) the phylogenetic analysis does not include all described species and for that reason it becomes unclear whether the analyzed strains are in fact a new species or one of the species not included in the phylogenetic analysis; (2) the genetic characterization does not identify unique sets of genes present in just these strains when compared to other species in the genus; and (3) it is unknown if the titan-like phenotype and morphological features detected are specific of these strains or whether other species in the genus, under the same growing conditions (microgravity with 5%CO2) would present similar phenotypes.

Other issues.

1. From the manuscript it is unclear how many species are currently described in the genus Naganishia. The introduction mentions 14 species (pp 52-55), and the results section indicates 18 species in the MycoBank, and 11 species in the CBS (pp 278-279). Yet, only 10 species were included in the phylogenetic analysis.

2. Microgravity and 5% CO2. The authors should indicate the reason to add the 5% CO2 to the growing conditions tested, and whether this was necessary to induce the titan-like phenotype when cells were kept at simulated microgravity ?
3. Methods section. 
3.1. WGS were not generated during this study as referred (pp 137-138). In Data availability the authors indicate that WGS have been deposited in GenBank but this must have been done in a previous publication.
3.2. “phylogenetic analyses based on individual genes mentioned above” (pp140), not clear what are the genes that the authors refer here. 
3.3. Missing in methods: the authors do not mention how homologous sequences were identified in all genomes, and how evolutionary models of evolution were inferred. I would recommend that the authors follow methodologies used in Liu et al 2015 (reference 2 in the manuscript) to describe and perform the phylogenetic analyses.

4. The manuscript if full of rather sloppy formatting: (1) Repeated sentences in the introduction (pp 57-61); (2) repeated numbering in the methods section; (3) repeated sentences in the methods (pp 137-143); repeated information in the results (pp 306-309); (4) wrong calling of supplemental information; (5) We can´t read data on figure 5; (6) discussion (pp 543-547). 
5. This sentence is not clear (pp: 72-74).

Author Response

Rev 2 comments:

The authors characterize four strains sampled from the International Space Station and propose that these are representatives of a new undescribed species of the genus Naganishia. Interestingly the authors show that these cells adopt a titan-like phenotype when grown under simulated microgravity combined with 5% CO2 conditions. This is a phenotype that has been previously associated to pathogenicity in Cryptococcus yeasts.

The carefully annotation of these new genomes and the detailed characterization of the titan-like phenotype in the cells morphology is novel and interesting.

Yet, the authors have not provided compelling information that these strains are in fact from a new species.

(1) the phylogenetic analysis does not include all described species and for that reason it becomes unclear whether the analyzed strains are in fact a new species or one of the species not included in the phylogenetic analysis;

Ans: We have included all Naganishia species mentioned in MycoBank or CBS databasein the phylogenetic analysis. As stated in the manuscript, the MycoBank database legitimately documented 19 Naganishia species, but the CBS database showed only 11 Naganishia species. Recently, yet another species Naganishia floricola published that is neither in MycoBank nor in CBS database. In total, 20 Naganishia ITS region sequences were retrieved from NCBI and the phylogenetic tree constructed (see modified Suppl Figure S1). However, sequences of seven genes used in the MLST analysis as per Liu et al [1] were available for only 10 Naganishia species on NCBI and a phylogenetic tree was constructed with C. neoformans as an outgroup. In addition, the individual phylogenetic trees for all these seven genes were also given in the supplemental figures (S1 to S7). ITS sequence was available for 19 species, LSU and SSU sequences were existing for 13 Naganishia species, CytB, PRB1, and TEF1 sequences were found only in 12 Naganishia species and finally RPB2 gene sequences were available for only 10 Naganishia species. Hence, we constructed MLST phylogenetic tree for only those 10 Naganishia species (see Fig 1). Therefore, authors are confident that they included all of the available Naganishia species in the phylogenetic analysis and found that the ISS strains were novel and not previously described.

(2) the genetic characterization does not identify unique sets of genes present in just these strains when compared to other species in the genus; and

Ans: As noted by the reviewer, we did not identify any unique sets of genes for N. tulchinskyi and will be carried out after seeing more genomes of Naganishia available. As on Nov 16, 2021, the NCBI database shows only 5 genomes of various Naganishia species plus our ISS genomes (https://www.ncbi.nlm.nih.gov/genome/browse#!/overview/naganishia). However, that is not required to delineate the novel species in yeast as per Liu et al. [1]. Authors have included clear phylogenetic evidence of the novel species (MLST analyses; see Fig 1; Supplemental Fig 1 to 7) as well as a holotype description (Section 3.2) consistent with previous description of novel yeast species [1].

(3) it is unknown if the titan-like phenotype and morphological features detected are specific of these strains or whether other species in the genus, under the same growing conditions (microgravity with 5%CO2) would present similar phenotypes.

Ans: The point raised by the reviewer is very interesting. Since the Naganishia species isolated from ISS are needed to test under simulated microgravity conditions, authors have restricted their analysis only to the ISS strains. In addition, WGS of ~1,000 other ISS strains did not reveal presence of any other Naganishia species other than the novel one. In the future, other members of the genus Naganishia will be tested for Titan-like cell morphology formation due to simulated microgravity.

Other issues.

  1. From the manuscript it is unclear how many species are currently described in the genus Naganishia. The introduction mentions 14 species (pp 52-55), and the results section indicates 18 species in the MycoBank, and 11 species in the CBS (pp 278-279). Yet, only 10 species were included in the phylogenetic analysis.

Ans: We have included all Naganishia species mentioned in MycoBank or CBS databasein the phylogenetic analysis. As stated in the manuscript, the MycoBank database legitimately documented 19 Naganishia species, but the CBS database showed only 11 Naganishia species. Recently, yet another species Naganishia floricola published that is neither in MycoBank nor in CBS database. In total, 20 Naganishia ITS region sequences were retrieved from NCBI and the phylogenetic tree constructed (see modified Suppl Figure S1). However, sequences of seven genes used in the MLST analysis as per Liu et al [1] were available for only 10 Naganishia species on NCBI and a phylogenetic tree was constructed with C. neoformans as an outgroup. In addition, the individual phylogenetic trees for all these seven genes were also given in the supplemental figures (S1 to S7). ITS sequence was available for 19 species, LSU and SSU sequences were existing for 13 Naganishia species, CytB, PRB1, and TEF1 sequences were found only in 12 Naganishia species and finally RPB2 gene sequences were available for only 10 Naganishia species. Hence, we constructed MLST phylogenetic tree for only those 10 Naganishia species (see Fig 1). Therefore, authors are confident that they included all of the available Naganishia species in the phylogenetic analysis and found that the ISS strains were novel and not previously described.

  1. Microgravity and 5% CO2. The authors should indicate the reason to add the 5% CO2 to the growing conditions tested, and whether this was necessary to induce the titan-like phenotype when cells were kept at simulated microgravity?

Ans: The following rationale is added in the modified manuscript.

  • The CO2 level of ISS habitat is ~10x more (~4,000 ppm) than at Earth conditions [2] and also Titan cells were observed under 5% CO2 level in neoformans [3]. Hence, morphological changes were characterized at 5% CO2 level.
  1. Methods section.

3.1. WGS were not generated during this study as referred (pp 137-138). In Data availability the authors indicate that WGS have been deposited in GenBank but this must have been done in a previous publication.

Ans: As noted by the reviewer, we have published the draft genomes in a Microbial Resource Announcement journal [4]. Information on DNA extraction and WGS generation were deleted. However, when we saw the phenotype difference under microgravity as well as phylogenetic novelty, in-depth genomic analysis was carried out. Since data availability need to be stated as per journal conditions, we were giving the accession number again. If the reviewer feels this is a repetition, we can remove.

3.2. “phylogenetic analyses based on individual genes mentioned above” (pp140), not clear what are the genes that the authors refer here.

Ans: Information about the individual genes tested (ITS, LSU, SSU, CYTB, RPB1, RPB2, TEF1) was added in the modified manuscript.

3.3. Missing in methods: the authors do not mention how homologous sequences were identified in all genomes, and how evolutionary models of evolution were inferred. I would recommend that the authors follow methodologies used in Liu et al 2015 (reference 2 in the manuscript) to describe and perform the phylogenetic analyses.

Ans: We followed Liu et al. 2015 (seven-gene strategy) to phylogenetically infer the species. The following were the methods described by Liu et al. and described in the methods section.

  • Phylogenetic analysis was carried out using seven genes: internal transcribed spacer (ITS) region rRNA gene, D1/D2 domain of large subunit (LSU or 26S) rRNA gene, small subunit (SSU or 18S) rRNA gene, and housekeeping genes including two subunits of RNA polymerase II (RPB1 and RPB2), translation elongation factor 1-α (TEF1), and cytochrome b (CYTB), which were used for differentiating Tremellomycetes species [1]. The respective gene sequences that were available on NCBI for different Naganishia species were included in the phylogenetic analysis. In addition, genomes of closely related members from Tremellomycetes (Goffeauzyma gastricus CBS 2288 (AF145323.1), Heterocephalacria arrabidensis CBS 8678 (AF444362.2), Piskurozyma cylindrica CBS 8680 (AF444360.1), Solicoccozyma aerius CBS155 (AF145324.1), Tremella mesenterica CBS 6973 (AF444433.1), were included with Cryptococcus neoformans CBS 132 (AF444326.1) as outgroup.
  • Initially, seven individual phylogenetic trees using ITS, LSU, SSU, CTYB, RPB1, RPB2, and TEF1 genes were generated and presented in Supplemental Figures 1 to 7. The ITS and LSU-based phylogenetic trees (Supplemental Figures 1 and 2) displayed two clades; two strains showed albida as the closest neighbor and the other two strains were affiliated to N. diffluens CBS 160, indicating ITS, and LSU genes might not be useful to differentiate Naganishia species. When tested individually, the phylogenetic analyses of the other 5 genes (SSU, CYTB, RPB1, RPB2, TEF1) formed a single group for all the four ISS strains. There were no significant variations in SSU gene sequences (Supplemental Figure 3) whereas other four genes (CYTB, RPB1, RPB2, and TEF1) invariably exhibited higher resolution in differentiating all the Naganishia species. All of these phylogenetic trees placing the ISS strains together and the closest neighbor was identified as N. diffluens. Subsequently, MLST was carried out by manually concatenating the seven genes (Figure 1). The MLST phylogenetic tree constructed showed that the four ISS strains clustered together and in the same clade with N. diffluens CBS 160T (Figure 1). The individual gene-based analyses and MLST-based tree further supported that the four ISS strains belong to the same species but closely related to N. diffluens. This suggested that the four ISS strains were novel species of the genus Naganishia. Since all four strains were isolated at different flight and sampling sessions and at various locations, they are not clonal but persisting in the ISS environment, and its ecological significance in the closed systems warrants further study. The genomes of four ISS strains were sequenced, draft genome assembled, annotated, and the results were published elsewhere [4]. The genome size was ~19.4 Mbp with GC content between 53-56%, similar to other members of the genus Naganishia.
  1. The manuscript if full of rather sloppy formatting:

(1) Repeated sentences in the introduction (pp 57-61);

Ans: Inadvertently placed and removed in this revision.

(2) repeated numbering in the methods section;

Ans: More care was taken in the revised submission.

(3) repeated sentences in the methods (pp 137-143);

Ans: Assume reviewer is mentioning about the following paragraph? Authors removed the repeated sentences and should read as below.

The WGS of all strains were retrieved from NCBI except for the four ISS strains which were generated during this study. Phylogenetic analyses based on individual genes (ITS, LSU, SSU, CYTB, RPB1, RPB2, TEF1) and MLST analysis with seven genes were carried out with all available Naganishia species. The individual gene sequences for all strains were first aligned separately using ClusatalW and for MLST, seven gene sequences for each strain were concatenated manually, aligned using ClusatalW, and generated Maximum Likelihood Tree using MEGA 7.0.26 [5].

repeated information in the results (pp 306-309);

Ans: Authors believe the following information given in the said lines are not repeated again any other place.

  • Since all four strains were isolated at different flight and sampling sessions and at various locations, they are not clonal but persisting in the ISS environment, and its ecological significance in the closed systems warrants further study.

(4) wrong calling of supplemental information;

Ans: Modified as per journal style. It should state as Figure S1, S2…..etc.

(5) We can´t read data on figure 5;

Ans: In order to increase the legibility of this figure, we have reduced the matrices to include only GOslim terms that are present in the proteome of the type species (IF6SW-B1T).

(6) discussion (pp 543-547).

Ans: Inadvertently placed texts are removed. Since Results and Discussion were combined, this section is removed in this version.

  1. This sentence is not clear (pp: 72-74).

Ans: Modified. The “bio” phrase was removed.

  • 1. Liu, X.Z.; Wang, Q.M.; Theelen, B.; Groenewald, M.; Bai, F.Y.; Boekhout, T. Phylogeny of tremellomycetous yeasts and related dimorphic and filamentous basidiomycetes reconstructed from multiple gene sequence analyses. Studies in mycology 2015, 81, 1-26, doi:10.1016/j.simyco.2015.08.001.
  • 2. Checinska Sielaff, A.; Urbaniak, C.; Mohan, G.B.M.; Stepanov, V.G.; Tran, Q.; Wood, J.M.; Minich, J.; McDonald, D.; Mayer, T.; Knight, R.; et al. Characterization of the total and viable bacterial and fungal communities associated with the International Space Station surfaces. Microbiome 2019, 7, 50, doi:10.1186/s40168-019-0666-x.
  • 3. Hommel, B.; Mukaremera, L.; Cordero, R.J.B.; Coelho, C.; Desjardins, C.A.; Sturny-Leclère, A.; Janbon, G.; Perfect, J.R.; Fraser, J.A.; Casadevall, A.; et al. Titan cells formation in Cryptococcus neoformans is finely tuned by environmental conditions and modulated by positive and negative genetic regulators. PLoS Pathog. 2018, 14, e1006982, doi:10.1371/journal.ppat.1006982.
  • 4. Bijlani, S.; Singh, N.K.; Mason, C.E.; Wang, C.C.C.; Venkateswaran, K. Draft Genome Sequences of Tremellomycetes Strains Isolated from the International Space Station. Microbiol Resour Announc 2020, 9, doi:10.1128/MRA.00504-20.
  • 5. Kumar, S.; Stecher, G.; Tamura, K. MEGA7: Molecular Evolutionary Genetics Analysis Version 7.0 for Bigger Datasets. Mol Biol Evol 2016, 33, 1870-1874, doi:10.1093/molbev/msw054.

Reviewer 3 Report

The authors carried out genetic characterization and description of a new species.

Sequence of housekeeping genes seem to support the assumption, that a new species was found.

My concern with presented phylogenetic analysis is the use of partial sequences found only in one copy per strains. From this data the dominant copy cannot be assumed. I strongly recommend to sequence the ITS region from each strains of N. tulchinskyi.

The LSU tree (Suppl. fig. 2) is incorrect (shows the ITS tree and refers to ITS sequences), therefore the conclusion “ITS, and LSU genes might not be useful to differentiate Naganishia species” is false. LSU for the isolated strains are the same (JAAZQA010000020.1, JAAZPV010000111.1, JAAZPY010000121.1), does not place them into different clades. The fourth strain (Naganishia IF1SW-F1) has only partial sequence, at least D1-D2 region should be sequenced.

Having the necessary sequences, the phylogenetic analysis should be revised.

Since SSU fragments are rather short, I recommend to omit this gene from the analysis.

The description of the proposed novel species in its current form is unacceptable for numerous reasons exemplified below.

The authority names are missing.

The standard phenotypic description of the novel species is missing. Please consult with the latest edition of the Yeast, a Taxonomic Study (Kurtzman et al., 2011) and recent descriptions of novel yeast species published by leading yeast research groups.

The single holotype (which is maintained in a metabolically inactive state) is not specified in the description. Isotype or ex-type cultures must be deposited in at least two recognised culture collections to ensure their availability for further research.

Fungal taxa and not strains have MycoBank numbers.

Technical comments:

N, diffluens misspelled 5 times (diffuens).

line 337: do you mean multilateral budding?

Discussion (lines 543-547) is missing (contains the instructions for authors…)

The article should be rewritten accordingly.

Author Response

Rev 3 comments:

The authors carried out genetic characterization and description of a new species.

Sequence of housekeeping genes seem to support the assumption, that a new species was found.

My concern with presented phylogenetic analysis is the use of partial sequences found only in one copy per strains. From this data the dominant copy cannot be assumed. I strongly recommend to sequence the ITS region from each strains of N. tulchinskyi.

Ans: In addition to the WGS, we performed amplicon sequencing of the ITS region for each of the 4 strains [1] and their accession numbers are given below. This information is added in the modified manuscript.

Naganishia IF6SW-B1 (KY218715.1); Naganishia IF1SW-F1 (KY218664.1); Naganishia IF7SW-B1 (KY218717.1); Naganishia IIF5SW-F1 (KY218695.1)

The LSU tree (Suppl. fig. 2) is incorrect (shows the ITS tree and refers to ITS sequences), therefore the conclusion “ITS, and LSU genes might not be useful to differentiate Naganishia species” is false. LSU for the isolated strains are the same (JAAZQA010000020.1, JAAZPV010000111.1, JAAZPY010000121.1), does not place them into different clades. The fourth strain (Naganishia IF1SW-F1) has only partial sequence, at least D1-D2 region should be sequenced.

Ans: Inadvertently inserted the ITS tree for LSU tree and this is a cut and paste error. Now the correct LSU tree is placed in the Suppl Fig S2.

Having the necessary sequences, the phylogenetic analysis should be revised.

Ans: Correct LSU figure inserted.

Since SSU fragments are rather short, I recommend to omit this gene from the analysis.

Ans: We have not seen any changes in the MLST tree with or without SSU. Authors wish to keep the tree with seven-genes as recommended by Liu et al (2015).

The description of the proposed novel species in its current form is unacceptable for numerous reasons exemplified below.

Ans: Modified as per the example.

The authority names are missing.

Ans: Added

The standard phenotypic description of the novel species is missing. Please consult with the latest edition of the Yeast, a Taxonomic Study (Kurtzman et al., 2011) and recent descriptions of novel yeast species published by leading yeast research groups.

Ans: Modified

The single holotype (which is maintained in a metabolically inactive state) is not specified in the description. Isotype or ex-type cultures must be deposited in at least two recognised culture collections to ensure their availability for further research.

Ans: Modified. The strain is deposited in NRRL and DSMZ culture collections.

Fungal taxa and not strains have MycoBank numbers.

Ans: MycoBank number is added.

Technical comments:

Ans:

N, diffluens misspelled 5 times (diffuens).

Ans: Corrected

line 337: do you mean multilateral budding?

Ans: Modified as “multilateral budding”

Discussion (lines 543-547) is missing (contains the instructions for authors…)

The article should be rewritten accordingly.

Ans: Removed. Since Results and Discussion was combined, this section is removed in this version.

  • 1. Checinska Sielaff, A.; Urbaniak, C.; Mohan, G.B.M.; Stepanov, V.G.; Tran, Q.; Wood, J.M.; Minich, J.; McDonald, D.; Mayer, T.; Knight, R.; et al. Characterization of the total and viable bacterial and fungal communities associated with the International Space Station surfaces. Microbiome 2019, 7, 50, doi:10.1186/s40168-019-0666-x.

Round 2

Author Response

Edits suggested by the reviewer 1 are incorporated into the document.

Point by point responses are provided below.

Line #258: Commented [M10]: why not 11 or 19, please explain it. Please point out whether or not the DNA sequences of these species originated from extype strains. 

  • Ans: Genomes of only 10 Naganishia species were available.

Line #259: Commented [M11]: Please provide the strain number.

  • Ans: Strain # CBS 132 is provided.

Reviewer 3 Report

The standard phenotypic description, which is a fundamental part of the description of a novel yeast species, is still very rudimentary, practically missing. Replacing cells by conidia (probably suggested by one of the reviewers) did not improve the description, on the contrary. What about the physiological and biochemical characters of the investigated strains? The NRRL and DSMZ accession numbers must be provided. Collection acronyms do not identify strains.

The D1-D2 region of 26S and ITS regions are also an essential part of a species description. The results presented are contradictory and therefore do not allow reliable sequences to be determined for these regions.

D1-D2 sequences of the amplicons (KY218715, KY218717, KY218664, KY218695) show significant (25) differences compared to WGS. Ten times higher than the interspecific variability of this clade.

ITS sequences (KY218717, KY218664, KY218695) are identical, do not differentiate the strains as shown in Suppl Figure 1 and on the concatenated tree.

ITS sequences of strain IF6SW-B1 (KY218715 and WGS) differ 40 in/dels and sst. I'm afraid that's not acceptable.

Author Response

Please see the attached file for authors response.

Round 3

Reviewer 3 Report

Physiological characters have been determined at least for the majority, but most probably for all currently accepted Naganishia species. So, there is no need to procure and characterise ex-type or other strains of all Naganishia species. (NB, species cannot be procured.)

The cited publications do not support omitting the physiological characterization. For example Aime et al. (2021) suggest the inclusion of assimilation panels in case of yeasts and all published examples of yeast descriptions listed by them include physiological characterisation. Liu et al. (2015) and Yurkov et al. (2021) do not suggest omitting physiological and biochemical characterisation of novel yeast species either. Raja et al. 2017 does not mention the word yeast.

If a strain is deposited in a culture collection as stated earlier by the Authors, the accession number must be available. Otherwise the deposition might be in progress.

I have never claimed ITS would be the gold standard, especially for yeasts, where the D1-D2 is the most frequently used region for identification. Sequences of the amplicons (KY218715, KY218717, KY218664, KY218695) do contain D1-D2 region (approx.: 610-1180bp) and show several differences comparing to those retrieved from WGS!